https://doi.org/10.1038/s41467-020-20548-8　　**OPEN**

# Engineering and elucidation of the lipoinitiation process in nonribosomal peptide biosynthesis

Lin Zhong [1,3], Xiaotong Diao [1,3], Na Zhang[1,3], Fengwei Li[1], Haibo Zhou[1], Hanna Chen[1], Xianping Bai[1], Xintong Ren [1], Youming Zhang[1✉], Dalei Wu [1,2✉] & Xiaoying Bian [1✉]

Nonribosomal peptide synthetases containing starter condensation domains direct the biosynthesis of nonribosomal lipopeptides, which generally exhibit wide bioactivities. The acyl chain has strong impacts on bioactivity and toxicity, but the lack of an in-depth understanding of starter condensation domain-mediated lipoinitiation limits the bioengineering of NRPSs to obtain novel derivatives with desired acyl chains. Here, we show that the acyl chains of the lipopeptides rhizomide, holrhizin, and glidobactin were modified by engineering the starter condensation domain, suggesting a workable approach to change the acyl chain. Based on the structure of the mutated starter condensation domain of rhizomide biosynthetic enzyme RzmA in complex with octanoyl-CoA and related point mutation experiments, we identify a set of residues responsible for the selectivity of substrate acyl chains and extend the acyl chains from acetyl to palmitoyl. Furthermore, we illustrate three possible conformational states of starter condensation domains during the reaction cycle of the lipoinitiation process. Our studies provide further insights into the mechanism of lipoinitiation and the engineering of nonribosomal peptide synthetases.

[1] Helmholtz International Lab for Anti-Infectives, Shandong University-Helmholtz Institute of Biotechnology, State Key Laboratory of Microbial Technology, Shandong University, Qingdao, Shandong 266237, China. [2] Suzhou Research Institute, Shandong University, Suzhou, Jiangsu 215123, China. [3] These authors contributed equally: Lin Zhong, Xiaotong Diao, Na Zhang. ✉email: zhangyouming@sdu.edu.cn; dlwu@sdu.edu.cn; bianxiaoying@sdu.edu.cn

Nonribosomal peptide synthetases (NRPSs) are multi-modular enzymes that catalyze the biosynthesis of diverse peptides with a wide variety of activities[1,2]. Each NRPS module, which typically includes a condensation (C) domain, an adenylation (A) domain, and a peptidyl carrier protein (PCP) domain, incorporates a single amino acid into a growing peptide chain, forming the peptidyl backbone by sequential condensations[3]. The nonribosomal lipopeptides represented by the last resort antibiotics daptomycin and polymyxin, featuring an acyl tail on the N-terminus of the linear or cyclic peptidyl backbone, are typical products of NRPSs[4–9]. Because the N-terminal acyls are highly diverse, ranging from short acetyl to long fatty acyl groups, we here mention the NRPS-derived peptide with an N-acylation as a nonribosomal lipopeptide for the purpose of a convenient description. The incorporation of an acyl into peptidyl backbone is usually implemented by a starter condensation (Cs) domain, normally located at the beginning of the initial NRPS module, via the formation of an amide bond to the α-amino group of the first amino acid residue, a process also known as lipoinitiation[10]. The acyl moiety is delivered to the Cs domain either by tethering to an acyl carrier protein (ACP), as in the cases of calcium-dependent antibiotics (CDAs) and lipopeptide A54145[11,12], or simply in a free acyl-CoA form, as in the cases of

surfactin and glidobactin[13,14] (Fig. 1a). The N-terminal acyl chains of lipopeptides strongly affect the biological activity and toxicity of the lipopeptides[4,15–17]. Thus, the engineering of NRPS machinery (especially the Cs domain) to produce lipopeptides with a desired acyl chain is of great interest, and its success depends heavily on our understanding of the specificity and mechanism of lipoinitiation.

The mechanism of the lipoinitiation reaction has been investigated by in vitro biochemical experiments using the dissected Cs domains or entire initiation modules (Cs-A-PCP), respectively, from surfactin synthetase SrfAA, glidobactin synthetase GlbF, CDA synthetase CdaPS1, and A54145 synthetase LptA, revealing the substrate specificity of Cs domains for fatty acyl moieties[11–14]. The crystal structure of the dissected Cs domain from CdaPS1 enabled researchers to propose an active-site tunnel for the accommodation of both "donor" and "acceptor" substrates[18]. Due to the low affinity of C domains for substrate analogs, a covalent mimic of an acceptor substrate was adopted to obtain the cocrystal structure of CdaPS1-Cs, which revealed the role of second histidine (of the "HHxxxDG" motif) in substrate positioning and the acceptor substrate specificity of the condensation reaction[19]. Although a number of NRPS structures involving the C domains have been solved as single-domain[18–20] or

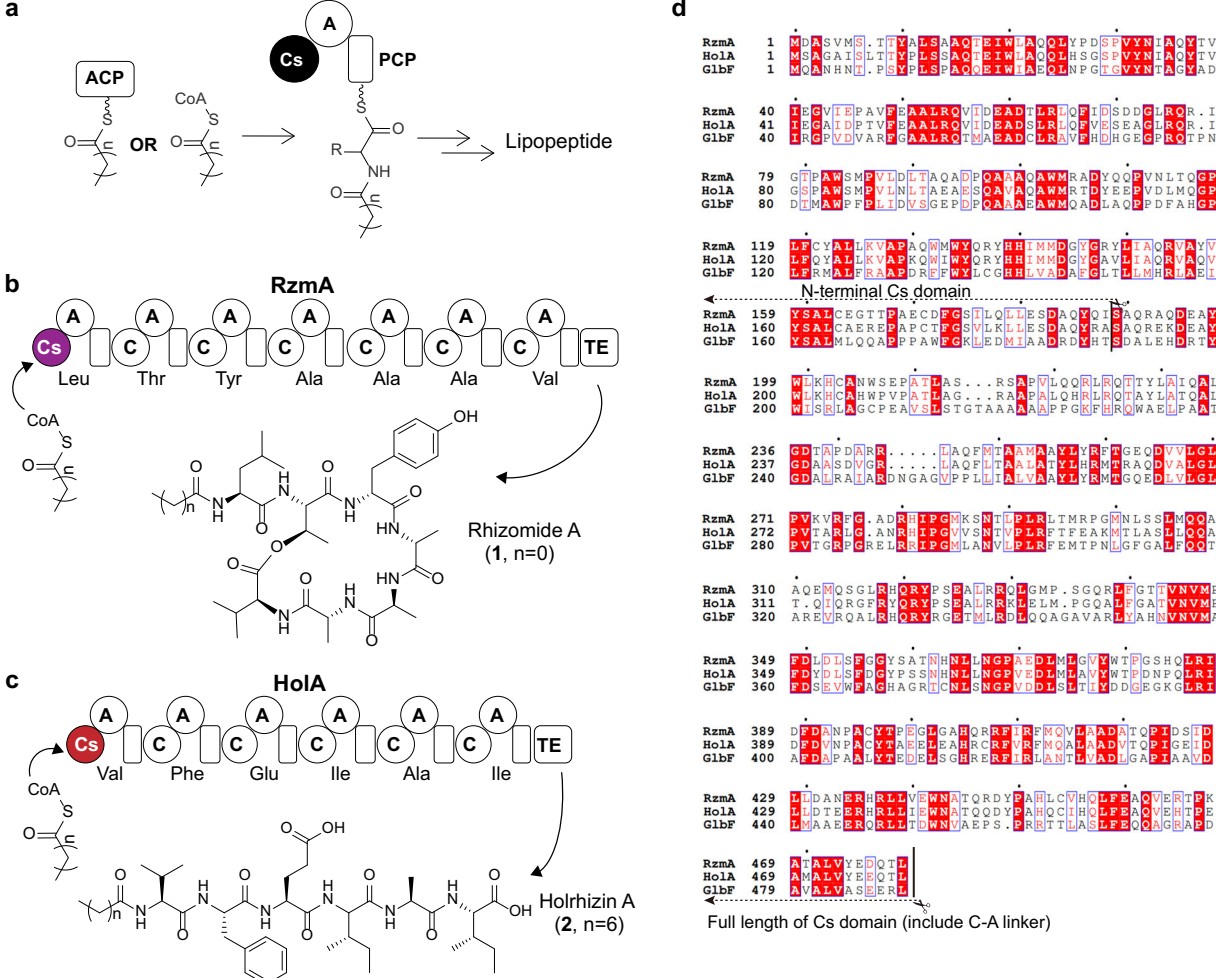

**Fig. 1 Biosynthesis of nonribosomal lipopeptides. a** The starter condensation (Cs) domain catalyzes the lipoinitiation of lipopeptide biosynthesis using acyl-ACP or acyl-CoA as a substrate. **b, c** The rhizomide and holrhizin biosynthetic pathways (RzmA and HolA) from *P. rhizoxinica*, and their main products rhizomide A (**1**) and holrhizin A (**2**), which bear a short acetyl and a medium octanoyl moieties, respectively. **d** Sequence alignment of RzmA-Cs, HolA-Cs, and GlbF-Cs. The swapping sites of the full-length Cs domain and subdomain (N-lobe) are labeled with a black line and a pair of scissors, respectively. A adenylation domain, ACP acyl carrier protein, C condensation domain, PCP peptidyl carrier protein, TE thioesterase domain.

multidomain (even modular) structures[21–28], CdaPS1-Cs remains the only lipoinitiation-conducting Cs domain with a reported structure. Moreover, very few C domain structures were solved in complex with substrates; in particular, none were solved with the acyl moiety. These setbacks obstruct the elucidation of the lipoinitiation mechanism and engineering of the acyl chain of lipopeptides.

Two types of lipopeptides, rhizomide A (**1**) and holrhizin A (**2**), with different acyl chains (Fig. 1b, c), were discovered in the bacterium *Paraburkholderia rhizoxinica* HKI 454 in our previous work[29–31]. Their synthetases possess Cs domains with a high sequence identity. However, the lengths of acyl chains they normally load are quite different (C2 vs C8), making this pair of lipopeptides an ideal model to investigate donor substrate specificity.

In this work, we successfully change the acyl chains of both rhizomide and holrhizin by swapping the Cs domain or mutation of a key amino acid residue, and obtain a Cs mutant with stronger binding to the acyl-CoA substrate. Then, we solve multiple cocrystal structures of RzmA-Cs mutants with donor and/or acceptor substrates. The co-complex with octanoyl-CoA (C8-CoA) reveals a binding pocket within RzmA-Cs to determine the selectivity of lipid chains, and donor substrate is extended up to palmitoyl-CoA (C16) from native acetyl-CoA (C2) through additional point mutations in vivo and in vitro. In addition to substrate specificity, these unliganded and co-complex structures of RzmA-Cs facilitate our understanding of other aspects of lipoinitiation, including the confirmed substrate positioning function of the second histidine in "HHxxxDG" motif for both "donor" and "acceptor" substrates, and more importantly the potential conformational changes that Cs domains (and likely also other C domains) would undertake throughout the condensation reaction cycle.

## Results

### Changing the acyl chains by Cs (sub)domain swapping.
Genome mining of two Cs domain-embedded cryptic NRPS biosynthetic gene clusters, *rzmA* and *holA*, in the bacterium *P. rhizoxinica* HKI 454 resulted in the discovery of two distinct types of lipopeptides, cyclic rhizomide A (**1**) containing an uncommon acetyl chain (C2) (Fig. 1b) and linear holrhizin A (**2**) harboring a medium octanoyl (C8) chain (Fig. 1c)[29–31]. Rhizomide A and holrhizin A differ in the length of acyl chains, suggesting that their Cs domains may be selective for different acyl substrates. Within the *rzmA* and *holA* biosynthetic gene clusters, no putative gene homologous to those encoding ACP or fatty acyl ligase (AL) was identified. Thus, we infer that the Cs domains of RzmA and HolA may catalyze lipoinitiation using acyl-CoAs as direct substrates, but the possibility of harnessing acyl-ACPs as direct substrates cannot be fully excluded, because the Cs domains might capture acyl-ACPs from the primary metabolism of bacteria. The in vitro biochemical reactions of dissected RzmA-Cs and HolA-Cs domains proved that they could catalyze the condensation between the acyl-CoAs (**3a** and **3d**) and corresponding first amino acid mimics, namely, aminoacyl-N-acetylcysteamine thioesters (aminoacyl-SNACs, **4** and **6**)[32] (Supplementary Fig. 1).

Although RzmA-Cs and HolA-Cs domains share a high sequence identity (73%, Fig. 1d), they showed totally different selections of donor acyl substrates in vitro and in vivo, furnishing a possible strategy to change the acyl chains of lipopeptides by the genetic engineering of Cs domains. We conducted full-length Cs domain (including C–A linker) and subdomain (N-lobe) swapping experiments in the *rzmA* and *holA* gene clusters by exchanging the coding regions of two Cs (sub)domains, and

transferred them into the heterologous host *Schlegelella brevitalea* strain DSM 7029 for functional expression (Figs. 1d, 2a–d)[33,34]. As expected, rhizomide derivative **1d** (Fig. 1b (n = 4), Fig. 2e) with an octanoyl (C8) chain and a holrhizin derivative **2a** (Fig. 1c (n = 0), Fig. 2f) with an acetyl (C2) chain were produced with decent yields, and their structures were confirmed by ultra-performance liquid chromatography coupled with high-resolution tandem mass spectrometry (UPLC–HRMS/MS, Supplementary Figs. 2 and 3) and nuclear magnetic resonance (NMR, Fig. 2e, f, Supplementary Tables 1–3 and Supplementary Figs. 11–20), respectively. The subdomain (N-terminal or N-lobe) swapping of the Cs domain also generated the expected products (**1d** and **2a**) but in a relatively lower yield and conversion ratio (Fig. 2c, d), suggesting that Cs domain swapping would be a feasible approach to change the acyl chains of lipopeptides.

To further test this approach, we also performed Cs (sub)domain swapping in GlbF of the glidobactin biosynthetic pathway in the original producer *S. brevitalea* DSM 7029[35]. The full-length GlbF-Cs domain was replaced by RzmA-Cs, HolA-Cs, as well as GlpC-Cs domain derived from the glidopeptin synthetase GlpC, which incorporates a decanoyl (C10) chain[29]. The exchange of HolA-Cs and Glp-Cs changed the original unsaturated 2(E),4(E) dodecadienoyl of glidobactin A to saturated octanoyl (C8) and decanoyl (C10) chains with improved yields (Fig. 3a, b, NMR: Supplementary Table 4, Supplementary Figs. 29–32), respectively. However, the change to RzmA-Cs failed to produce any products, possible due to the acceptor specificity of the RzmA-Cs domain and/or the donor specificity of the downstream C domains. The acceptor specificities of the three Cs domains were tested in vitro (Fig. 3c, d), and the RzmA-Cs and HolA-Cs showed relatively broad specificities to the tested acceptor substrates including Thr, indicating that the acceptor specificity of the RzmA-Cs domain failed to affect the swapped glidobactin product. Thus, the reason for the failure could be donor specificity of the downstream C domain. The relatively broad specificities of RzmA-Cs and HolA-Cs to acceptor substrates explain why the aforementioned Cs domain swapping led to the successful production of acyl-changed rhizomide, holrhizin, and glidobactin derivatives, and support the very recent study that novel nonribosomal peptides can be generated by the substitution of A domains alone[36]. Subdomain (N-lobe) swapping of GlbF-Cs with that of GlpC-Cs was also conducted, but no expected compounds were detected. Combined with the swapping in RzmA and HolA (Fig. 2), we propose that the full-length Cs domain swapping could be a worthy choice to modify the acyl chains of lipopeptides, at least for these three lipopeptides.

### A key site for controlling of the length of the acyl chain.
Next, we tried to determine how the highly similar RzmA-Cs and HolA-Cs domains recognize and select acyl-CoAs with different lengths. The alignment of protein sequences and the construction of a 3D homology model using the structure of CdaPS1-Cs[18] as a template allowed us to examine those residues that are not conserved in two proteins along the model active-site tunnel (Supplementary Fig. 4). Among a small group of candidate residues, R148 of RzmA (corresponding to A149 of HolA and M165 of CdaPS1) caught our attention. This residue not only sits near the proposed binding pocket for the 2,3-epoxyhexanoyl chain of CDA in the CdaPS1 structure[18], but also fits well with the simple logic that a longer side-chain of pocket residue would correlate with a shorter substrate chain, and vice versa (i.e., R148 with C2, M165 with C6, and A149 with C8) (Fig. 4a). Therefore, we generated a series of point mutations at this position in *rzmA* and *holA* to test whether the selectivity of acyl chains would be changed in vivo (Fig. 4b). We found that compared with

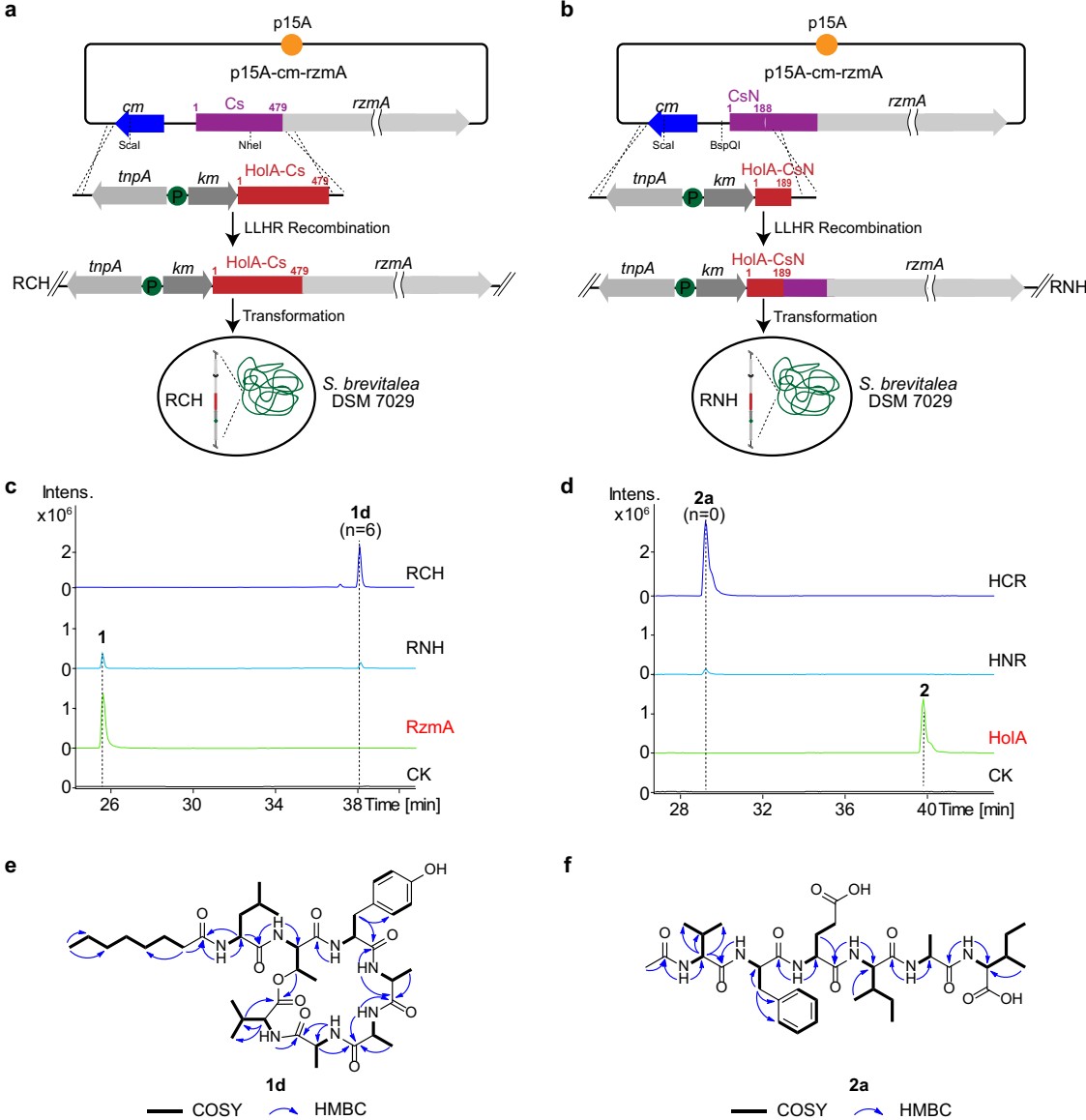

**Fig. 2 Exchanging the acyl chains of nonribosomal lipopeptides by Cs (sub)domain swapping. a, b** Swapping strategy for full-length Cs domain (RCH, **a**) and N-terminal Cs domain (RNH, **b**) of RzmA with those of HolA using recombineering in *E. coli*[46]. The resulting constructs (RCH and RNH) were transformed into *S. brevitalea* DSM 7029 for heterologous expression. The Cs (sub)domain swapping of HolA with those of RzmA (HCR and HNR) was conducted in the same way. **c, d** Comparative production analysis of full-length and N-terminal Cs domain swapping of RzmA (**c**) and HolA (**d**) showed an exchange of the acyl chains with different efficiencies. Extracted ion chromatograms (EICs) at $m/z = 732.391$ ([M + H]+, **1**) and 816.483 ([M + H]+, **1d**) for (**c**), and at $m/z = 817.503$ ([M + H]+, **2**) and 733.412 ([M + H]+, **2a**) for (**d**). CK represents the wild type of *S. brevitalea* DSM 7029. **e, f** Structures and key COSY and HMBC correlations of **1d** and **2a**. tnpA transposase gene, km kanamycin resistance gene, cm chloramphenicol resistance gene, P promotor. LLHR, linear plus linear homologous recombination. Full information was given in "Methods" section and Supplementary files.

wild-type (WT) RzmA, mutations at R148 indeed altered the production profile of rhizomide in vivo (Fig. 4c). As the residue side chains became shorter (R → M → L → V → A → G), the acyl chains of rhizomides grew longer from C2 to C8 or C10 (Fig. 4c). The in vitro substrate competition experiments of excised RzmA-Cs domain variants containing point mutations at R148 using a mixture of equal concentrations of C2-C18 acyl-CoAs as substrates also showed similar alteration of the substrate bias of each mutant (Supplementary Fig. 1e), basically in coherence with our in vivo experiments. A similar trend, but in the reverse direction could be seen for the HolA A149 mutations, in which longer residue side chains led to shorter acyl chains (Fig. 4d). These synchronized changes showed that the length of the side-chain of this key amino acid (R148 in RzmA-Cs, A149 in HolA-Cs) might

control the length of acyl chains. The side-chain may act as an entry gate to a potential pocket. Therefore, truncating the side-chain of this residue could give a correspondingly deeper space for the extension of longer acyl-CoAs. These results clearly highlight the important role of residues at this position of Cs domains, in defining the length of acyl chains during lipopeptide biosynthesis.

**Co-complex structure-guided engineering of the Cs domain.** In addition to the above finding of RzmA R148 (HolA A149), there are still many unanswered questions about lipoinitiation, such as whether more residues are involved in the recognition of substrate acyl chains and whether the production of lipopeptides

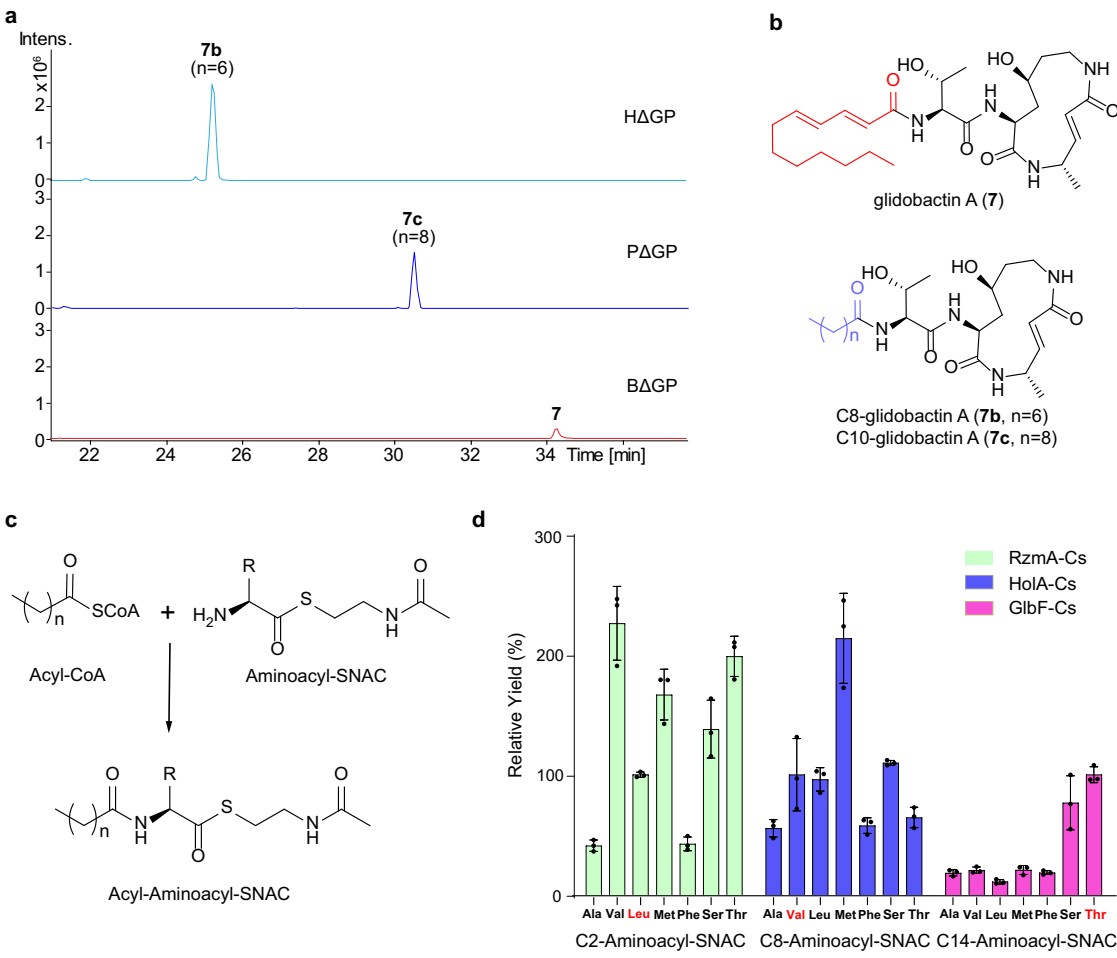

**Fig. 3 Cs domain swapping experiments on the glidobactin synthase GlbF and in vitro assay of Cs specificity for acceptor substrates. a**, **b** Production analysis of full-length domain swapping of GlbF-Cs (BΔGP) with HolA-Cs (HΔGP) and GlpC-Cs (PΔGP) in *S. brevitalea* DSM 7029 showed an exchange of the acyl chains with different yields. EICs at $m/z = 521.333$ ([M + H]$^+$, **7**) for 7029BΔGP, and 469.3021 ([M + H]$^+$, **7b**) for 7029HΔGP, and at $m/z = 497.333$ ([M + H]$^+$, **7c**) for 7029PΔGP. **c**, **d** In vitro assay of Cs specificity for acceptor substrates. The final products were quantified by comparison with the EIC peak area of native SNACs (100%), i.e., for RzmA-Cs, every set of tests was quantified with products of L-Leu-SNAC. The native SNACs are marked in red. Experiments were conducted in triplicates ($n = 3$), data are presented as mean values +/− SD, and detailed information is given in the "Methods" section.

with longer chains (>C8) is also possible by rational engineering. For this purpose, we sought structural information on these proteins. Despite our successful purification of the dissected Cs domains of both RzmA and HolA, we were able to obtain crystals only for RzmA-Cs, and solved its structure at 2.3 Å in the *P*1 space group (Supplementary Table 5). Similar to other C domain structures reported previously[18,20,25,26], RzmA-Cs adopts a "V-shaped" configuration and consists of two subdomains (N- and C-lobes) each representing the chloramphenicol acetyltransferase (CAT) fold (Fig. 5a). Interestingly, in each asymmetric unit there are four molecules of RzmA-Cs with two different conformations (Fig. 5b). The overall structures of the two conformations are quite similar (RMSD = 1.52 Å for Cα), especially when the two lobes are compared separately (RMSD = 0.68 Å and 1.30 Å for Cα of the N- and C-lobes). The major difference comes from the relative positions of the two lobes (i.e., opening angles of the "V" as shown in Fig. 5c). It has been known that the "openness" of C domains can vary in different NRPSs (Fig. 5d)[18]. However, it is quite unique that here the RzmA-Cs presents two conformations at the same time, emphasizing its intrinsic flexibility before engaging the substrates.

To delineate the interactions between the Cs domain and acyl-CoA, we wanted to solve their co-complex structure. After our

initial attempts to cocrystallize WT RzmA-Cs with acyl-CoAs of various lengths failed, we noticed that the R148A mutant protein showed strong binding to C8-CoA, as evidenced by the high ΔTm in protein thermal shift assays (Supplementary Fig. 5) and a $K_d$ value of ~300 nM measured using isothermal titration calorimetry (ITC) (Fig. 6a). Thus we selected this mutant for crystallization, and successfully obtained both unbound and C8-CoA-bound structures (Fig. 6b and Supplementary Table 5). The unbound R148A structure is mostly identical to that of WT RzmA-Cs in two conformations (Supplementary Fig. 6a). However, the C8-CoA-bound R148A structure shows only one conformation (in the $P2_12_12_1$ space group), which is "tighter" than both conformations of the unbound structure (Fig. 6c), indicating that the two lobes of RzmA-Cs may close up to bind acyl-CoA substrates.

This co-complex structure revealed the detailed interactions between the RzmA-Cs R148A mutant and C8-CoA (Fig. 6b–d). The CoA portion binds at the "donor" site of Cs with its loading arm inserting into the active-site tunnel formed by two lobes, while the acyl chain turns nearly 90° and extends into the N-lobe as the R148A mutation opens up a pocket between the α5 helix (where the R148 residue is located) and the β sheet (Fig. 6d). Interestingly, the ε nitrogen of H140 (the second histidine of the

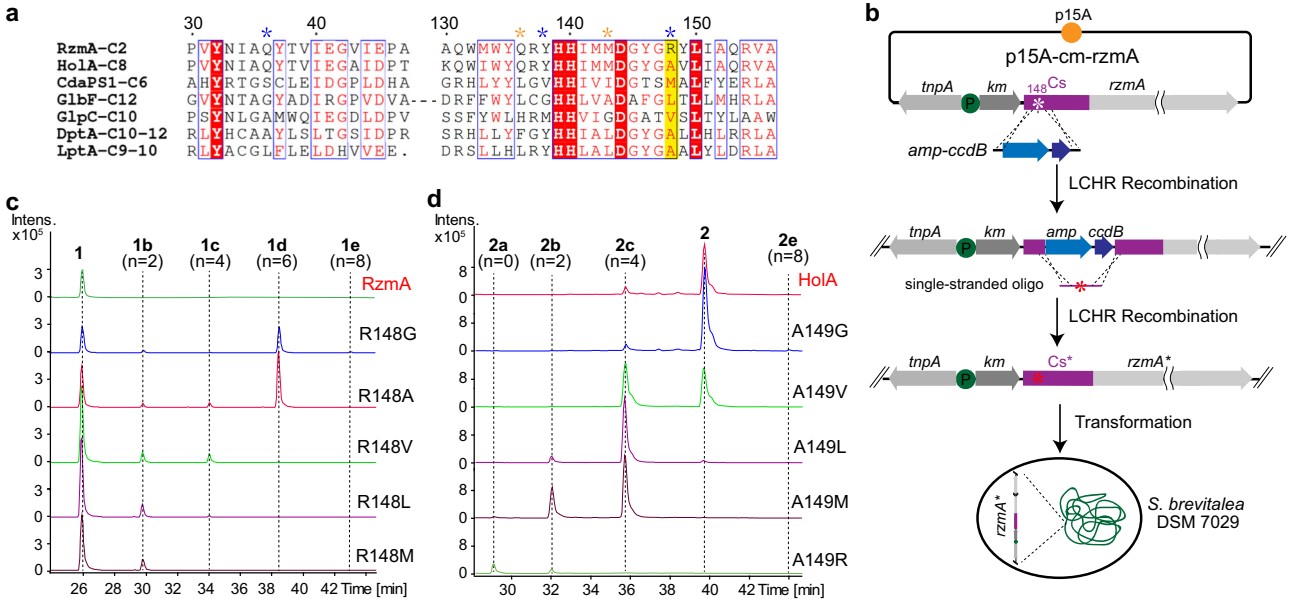

**Fig. 4 Point mutations of a key residue in the RzmA-Cs and HolA-Cs domains. a** Truncated sequence alignment of RzmA-Cs and HolA-Cs domains with other Cs domains loading different acyls. The asterisk-labeled residues represent key sites mentioned in the text for acyl substrate specificity (for details see Supplementary Fig. 4a). **b** The in vivo point mutation strategy of the *rzmA*-containing plasmid using CcdB counterselection recombineering[50, 51] The point mutations in *holA* were also performed in the same way. **c, d** Comparative production analysis of point mutations of RzmA R148 (**c**) and HolA A149 (**d**) to different residues in vivo showed gradient changes of the acyl chains of rhizomide and holrhizin, respectively. EICs at $m/z = 732.391$ (**1**), 760.421 (**1b**), 788.452 (**1c**), 816.483 (**1d**), and 844.513 (**1e**) for (**c**), and at $m/z = 733.412$ (**2a**), 761.441 (**2b**), 789.472 (**2c**), 817.503 (**2**), and 845.537 (**2e**) for (**d**). Wild-type RzmA and HolA expressed in *S. brevitalea* DSM 7029 are labeled in red. LCHR linear plus circular homologous recombination, amp-ccdB cassette of ampicillin resistance gene and CcdB toxin genes; tnpA and km are same as in Fig. 2. Full information was given in "Methods" section and Supplementary files.

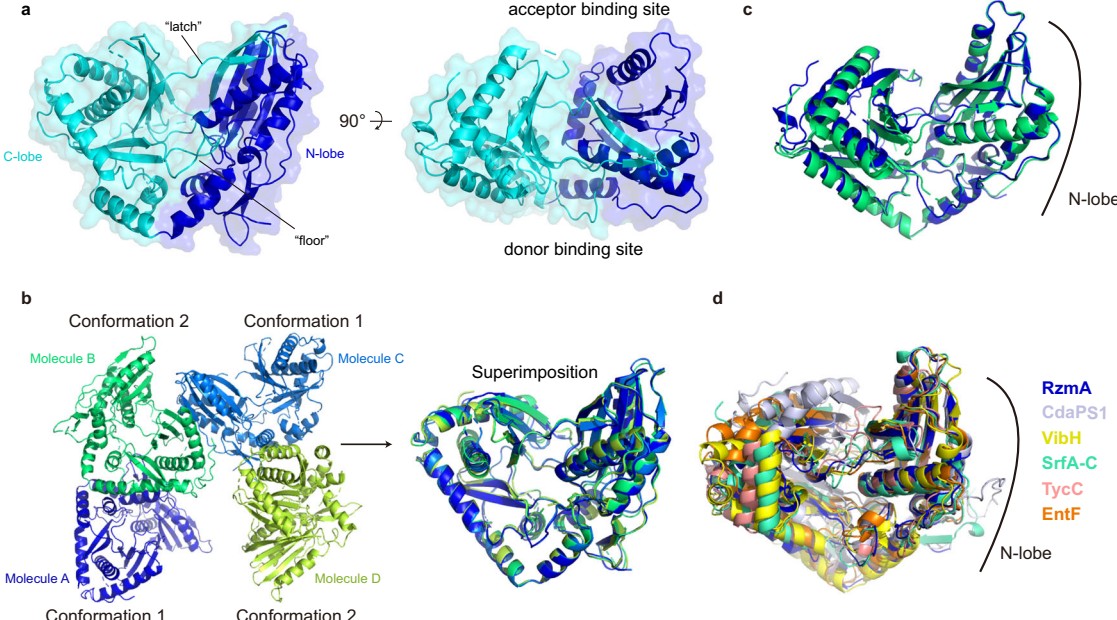

**Fig. 5 The overall structure of the RzmA-Cs domain. a** RzmA-Cs consists of N-lobe (blue) and C-lobe (cycan), and the C-lobe has two regions expanding to N-lobe, named as "latch" (residues 349–369) and "floor" (residues 274–287)[39]. **b** Four molecules of RzmA-Cs in each asymmetric unit with two conformations. **c** Comparison of two conformations of RzmA-Cs by overlaying the N-lobe. **d** Superimposition of C domains in six NRPSs, RzmA (PDB: 7C1H), CdaPS1 (PDB: 4JN5), VibH (PDB: 1L5A), SrfA-C (PDB: 2VSQ), TycC (PDB: 2JGP), and EntF (PDB: 5JA1), by their N-lobes.

"HHxxxDG" motif) is relatively close to the acyl group of the donor substrate (3.6 Å), suggesting that the substrate positioning function of this histidine applies not only to the acceptor substrate as inferred previously[19] but also to the donor substrate (the C8-CoA in this case). In addition, this structure helps us understand the conservation of G145 (within the "HHxxxDG" motif), because at this position in the C domain (starting point of the α5 helix), the residue has to sacrifice its side-chain to make room for proper binding and positioning of the donor substrates (Fig. 6d).

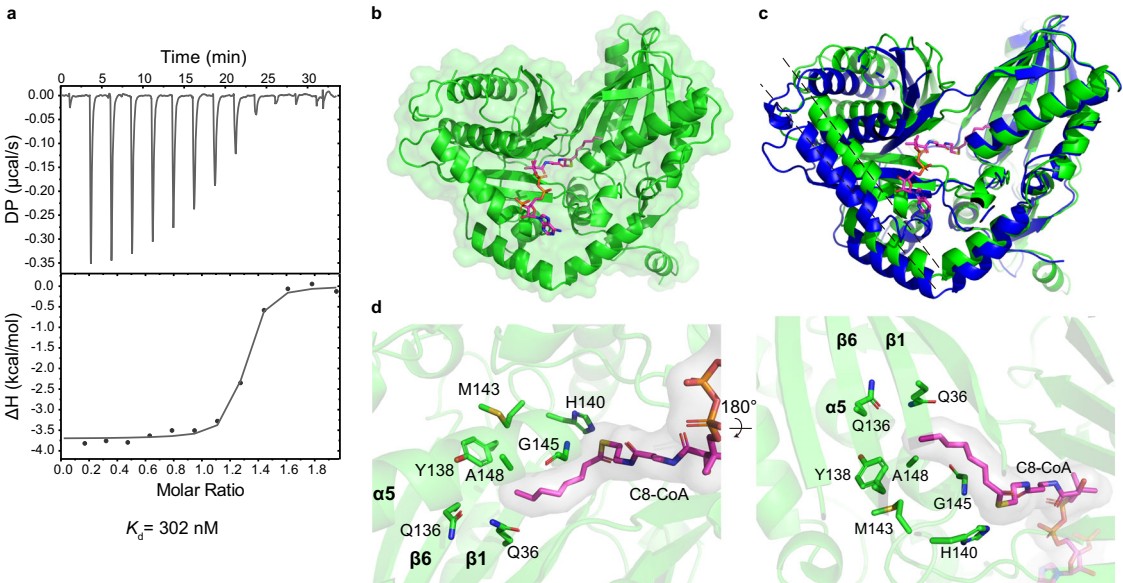

**Fig. 6 The co-complex structure of RzmA-Cs R148A with C8-CoA. a** Binding affinity measurement of C8-CoA to RzmA-Cs R148A using ITC. **b** Overall co-complex structure of R148A with C8-CoA. **c** Superimposition of R148A in complex with C8-CoA (green) and R148A unliganded (blue) by their N-lobes, highlighting the conformational changes of Cs domains during substrate binding. **d** The enlarged views of C8-CoA binding site. All the residues in sticks mode show their side chains only, except for the G145 which does not have a side-chain.

As mutation of the pocket "entry gate" residue R148 to glycine resulted in C8-rhizomide (**1d**) as the main product together with a tiny portion of C10-rhizomide (**1e**) (Figs. 4a, c and 7a), this mutation was chosen for further combinational mutations to explore the possibility of extending the length of the acyl chain on rhizomide. We first checked the residues near the tip of the C8 chain in the structure for mutation candidates to expand the binding pocket (Fig. 6d). Four residues, Q36 on the β1 strand, Q136 and Y138 on the β6 strand, and M143 on the loop between β6 and α5, were selected for this purpose. Mutations of these residues to Ala in combination with R148G clearly increased the length of the acyl chain incorporated into the peptidyl backbone both in vivo and in vitro (Fig. 7a, b). Compared with the mutation R148G alone, the additional mutations Q136A and M143A retained similar specificity for donor substrates, suggesting that Q136 and M143 may not significantly affect the length of the acyl chain. On the other hand, the additional mutations Y138A and Q36G changed the substrate preferences to longer acyl chains (C10-C12). However, the Q36G mutation markedly decreased the yields of rhizomide derivatives in vivo; thus, Y138 could be a superior mutation site for extension of acyl chains. The combined mutations Q136A/Y138A/R148G, Y138A/M143A/R148G, and Q136A/Y138A/M143A/R148G further shifted the preferences to C12-C16 fatty acyl-CoAs (Fig. 7a, b). The structures of rhizomide derivatives (**1b**–**1h**) containing different acyls were identified by UPLC–HRMS/MS and NMR analysis (Supplementary Fig. 2, Supplementary Tables 1 and 2, Supplementary Figs. 21–28), confirming substrate alteration in vivo. The triple mutant Y138A/M143A/R148G showed a large change in the specificity of the RzmA-Cs domain for acyl chains from short C2 to long C14, up to C16, with no marked reduction in product yields in vitro and in vivo (Fig. 7a, b), thus probably providing the optimal mutation sites for the extension of acyl chains in lipopeptides. These results correlate well with our hypothesis that rational engineering by mutation of Cs residues to reshape the substrate-binding pocket of acyl chains can lead to the production of lipopeptides with modified or even desired acyl chains.

**The mechanism of the lipoinitiation reaction**. To elucidate the lipoinitiation mechanism, we aimed to obtain the co-complex structure of RzmA-Cs with both donor (acyl-CoA) and acceptor (amino acid) substrates. To simplify the crystallization process, we utilized the substrate mimic L-Leu-SNAC[32] as in the above in vitro assays instead of the real substrate L-Leu loaded on the PCP domain from the same initiation module of RzmA (Fig. 1a). However, our initial crystallization attempts with WT or R148A mutant proteins all failed, possibly due to the fast reaction of the two substrates. To diminish the reaction rate, we made an additional mutation on the second histidine of the "HHxxxDG" motif[37] of RzmA-Cs R148A to create the double mutant H140V/R148A, which finally enabled us to obtain several crystals in the absence or presence of substrates (Supplementary Table 5).

We then solved and analyzed these H140V/R148A structures. The overall structure of unbound H140V/R148A is almost identical to the structures of WT and R148A in both conformations (Supplementary Fig. 6a). Then, we compared the C8-CoA-bound structures of H140V/R148A and R148A. Although their overall structures are quite similar (Supplementary Fig. 6b), we noticed that the electron densities surrounding the C8 acyl chain (but not the CoA portion) are much worse in the H140V/R148A structure than in the R148A structure (Supplementary Fig. 6c, d). These results indicate that the acyl chain of the substrate in the H140V/R148A structure is more flexible with H140 mutated to valine, again confirming the role of H140 in positioning the "donor" acyl-CoAs.

Next, we analyzed the structure of H140V/R148A in co-complex with two substrates. Its overall structure is almost identical to that of H140V/R148A bound to only C8-CoA (Fig. 8a, Supplementary Fig. 7a), despite a small conformational difference in C8-CoA itself (Supplementary Fig. 7b). In addition, the electron densities around both C8-CoA and L-Leu-SNAC are clearly visible (Supplementary Fig. 7c), making us confidently pinpoint how the Cs residues within the active site interact with both substrates. In particular, by mutating the key residue H140 back from V140 in the structure, we were able to show that its side-chain points right in the middle of two substrates, with a

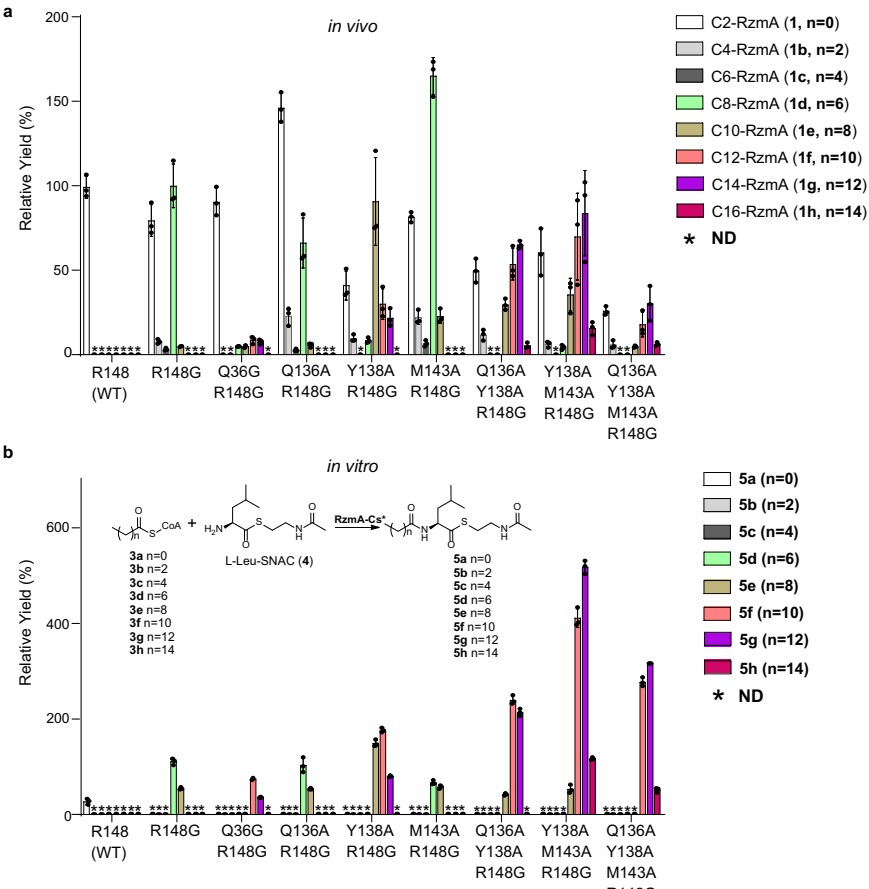

**Fig. 7 Structure-guided in-depth engineering of the Cs domain of rhizomide. a** The relative yields of rhizomide derivatives (**1**, **1b**–**1h**) with different acyl chains (C2-C16) in *S. brevitalea* DSM 7029 harboring the corresponding point-mutated *rzmA*. The yield of **1d** (C8-RzmA) in RzmA-R148G was quantified as a reference (100%) for the peak areas of EICs. Related chemical structures of rhizomide derivatives were given in Fig. 1b in which the '*n*' represents the number of $CH_2$ group. **b** The relative yields of acyl-Leu-SNAC (**5a**–**5h**) in the in vitro acyl-CoAs substrate competition experiment of RzmA-Cs variants. The yield of **5d** (C8-Leu-SNAC) in RzmA-Cs R148G was quantified as a reference (100%) for the peak areas of EICs. The products that were not detected (ND) by UPLC-MS are marked with asterisks. Experiments were conducted in triplicates (*n* = 3), data are presented as mean values +/− SD. The source data is provided in the Source data file.

distance of <4 Å to the reactive atoms from both sides (i.e., the "donor" carbonyl group and the "acceptor" α-amino group as shown in Fig. 8b). This structure, along with other structures described above or previously[19], undoubtedly indicates the key function of the second histidine at the "HHxxxDG" motif of Cs domains in positioning both substrates.

Finally, to depict how RzmA-Cs recognizes amino acids on the "acceptor" side, we zoomed into the binding site of L-Leu-SNAC in our structure (Supplementary Fig. 7d), and compared it with that of the chemical probe (as a substrate mimic) covalently bound to the Cs domain of the CdaPS1 E17C mutant (Fig. 8c)[19]. In the RzmA-Cs structure, three residues, P271, S287, and T289 (corresponding to P292, S309, and R311 in CdaPS1) near the side-chain of L-Leu may participate in substrate recognition (Fig. 8c). Interestingly, these three RzmA residues are fully conserved in HolA (Supplementary Fig. 4a), whose Cs domain accepts L-Val instead of L-Leu in the holrhizin biosynthetic pathway (Fig. 1c). Therefore, it was not surprising that RzmA-Cs could incorporate L-Val as the substrate both in vivo (Fig. 2c, d) and in vitro (Fig. 3c, d). In addition, the sequence alignment of several other Cs domains with the same or different substrates revealed no strict rule of conservation for the above three residues, except that the vast majority of them are short-chain residues (Supplementary Fig. 4a). The above in vitro experiments

with the RzmA-Cs, HolA-Cs, and GlbF-Cs domains also showed relatively wide specificities for acceptor substrates (Fig. 3c, d). All the above results are consistent with the fact that Cs domains can accommodate a certain degree of variation in the "acceptor" substrates, while the neighboring A domains are more responsible for substrate selectivity. This finding also supports the Cs domain swapping strategy to change the acyl chains (Figs. 2c, d, 3a, b), and the NRPS module swapping strategy at the C–A linker region for the combinatorial biosynthesis of nonribosomal peptides[36,38].

**Conformational changes in the reaction cycle.** In addition to the above three structures of the RzmA-Cs H140V/R148A mutant in either unbound or substrate-bound forms, we obtained a fourth ligand-free structure (Supplementary Table 5) derived from crystals that grew in drops containing H140V/R148A proteins and two substrates. However, these crystals usually appeared with a unique morphology after 2 weeks, much later than those in the co-complex with both substrates, which grew under the same conditions but took only 1–2 days (Supplementary Fig. 8). Interestingly, unlike the other unbound structures (WT, R148A, or H140V/R148A), which showed two conformations in the *P*1 space group, this structure of H140V/R148A was solved with only one conformation in the $P2_12_12_1$ space group (Fig. 9a).

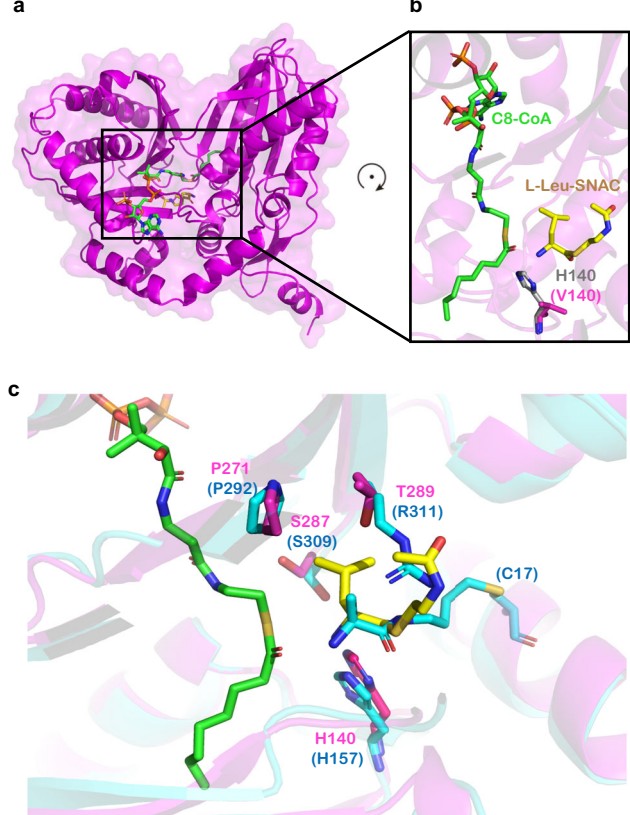

**Fig. 8 Interactions between RzmA-Cs and both substrates. a** Overall structure of the H140V/R148A mutant of RzmA-Cs in co-complex with C8-CoA (green) and L-Leu-SNAC (yellow). **b** An enlarged view around the two substrates. Valine 140 (magenta) is mutated back to histidine (white) to show its spatial position relative to the substrates. **c** Comparison of the L-Leu-SNAC binding site in RzmA-Cs with that of the covalent substrate mimic in the CdaPS1-Cs E17C mutant (PDB code 5DU9). Residues near the "acceptor" substrates in RzmA and CdaPS1 are labeled in magenta and blue, respectively.

Moreover, when comparing the structure of this ligand-free H140V/R148A with those of other substrate-unbound or substrate-bound forms (Supplementary Fig. 9a, b), we found that its overall conformation is very similar to that of substrate-bound H140V/R148A, with the majority of differences lying in the so-called "latch" and "floor" regions (Fig. 9b)[39]. This comparison indicated that the overturn of these two regions (especially the "floor" loop) dramatically increased the size of the active-site tunnel and expelled the C8-CoA substrate from the Cs domain (Supplementary Fig. 9c).

These above findings all point to the conclusion that we probably captured a "product-released" form of RzmA-Cs H140V/R148A. If this idea is correct, we should be able to detect the released product catalyzed by H140V/R148A in vitro. Therefore, we set up a testing system with a longer reaction time, and indeed detected the condensation product **5d** after a 300-min incubation of H140V/R148A at 30 °C, but in an extremely low yield compared with that of R148A after 100-min incubation (Fig. 9c). Thus, the H140V mutation did not completely abolish the condensation activity, correlating well with its role in substrate positioning[12,19]. Based on these H140V/R148A structures and related functional assays, we propose three potential conformational states (i.e., "unbound-bound-released") within the lipoinitiation reaction cycle catalyzed by Cs domains (Fig. 9d, also illustrated in Supplementary Video 1). The major

conformational changes during this cycle reflect the relative positions of two lobes, in concert with the binding of substrates and release of products. Notably, the "latch" and "floor" regions play essential roles in mediating and may also responding to conformational changes, as they are both crossover structures stretching from the C-lobe to the N-lobe, and their positions together determine the shape of the active-site tunnel (Supplementary Fig. 9c). Upon substrate binding, the two lobes of the Cs domain may close up to form proper interactions with the substrates, while a number of residues in the "latch" and "floor" regions may flip their side chains toward the active site to shrink the tunnel. After the condensation reaction, the Cs domain may open up slightly, accompanied by dramatic movement of the "floor" loop up against the binding position of CoA, possibly forcing it out of the "donor" site. Finally, the two lobes may open more, and the "floor" loop may move back down, from the "released" to the "unbound" position for a new cycle. This reaction cycle could also apply to other C domains, since their structures and functions are both closely related to those of Cs domains.

## Discussion

The lipopeptide natural products generated by the NRPSs containing Cs domains exhibit a wide range of bioactivities. These include antibiotics, anticancer drug candidates, and biosurfactants[4–7]. Some bacterial lipopeptides play roles in antagonism toward other (micro)organisms, motility, and attachment to surfaces[9]. Their biological activity in general depends on the interaction of their amphipathic structure with the cell membranes of target organisms[40]. Changing the acyl chain can balance the bioactivity and toxicity of lipopeptides, as shown in the development of daptomycin[4,15]. Several strategies have been harnessed to generate lipopeptide derivatives with different lipid groups for bioactivity evaluation. The classical semisynthetic method involves the deacylation of the native acyl chain by acylases to generate the peptide nucleus and subsequent reacylation with a desired acyl group by chemical reaction, which requires burdensome screening of efficient acylases and cumbersome (de)protection of other polar groups[15,16]. The genetic engineering method to change the acyl chain often relies on supplementing with alternate donor substrates based on manipulation of the lipid biosynthetic pathway or mutasynthesis. Directed mutagenesis of the active site of the β-ketoacyl-ACP synthase FabF3 of *Streptomyces coelicolor*, an enzyme in the biosynthesis of the CDA fatty acid moiety (C6), led to successful generation of two novel CDA derivatives with truncated (C4) acyl side chains[41]. Disruption of the biosynthetic gene for the native acyl chain of pneumocandin combined with the feeding of alternative side-chain precursors yielded four new pneumocandin congeners with straight C14-C16 side chains[42]. However, the yields were low, as the tolerance of the Cs domain for donor and acceptor substrates is limited.

Here, we changed the acyl chains of nonribosomal lipopeptides by bioengineering Cs domain in two ways, Cs domain swapping and point mutation. The recently identified exchange units (XUs) assembled at the fusion point located in the C–A linker region and the exchange unit condensation domain (XUC) fused at the point between the N-terminal and C-terminal subdomain of the C domain enabled the production of novel nonribosomal peptides without a significant decrease in yield[38,43]. We exchanged full-length Cs domains including the Cs-A linker or N-terminal subdomains (N-lobe), to alter the acyl chain length of lipopeptides. The full-length Cs domain exchange provided higher product yields than the N-terminal subdomain swapping method. The relatively broad specificities of the Cs domain for acceptor

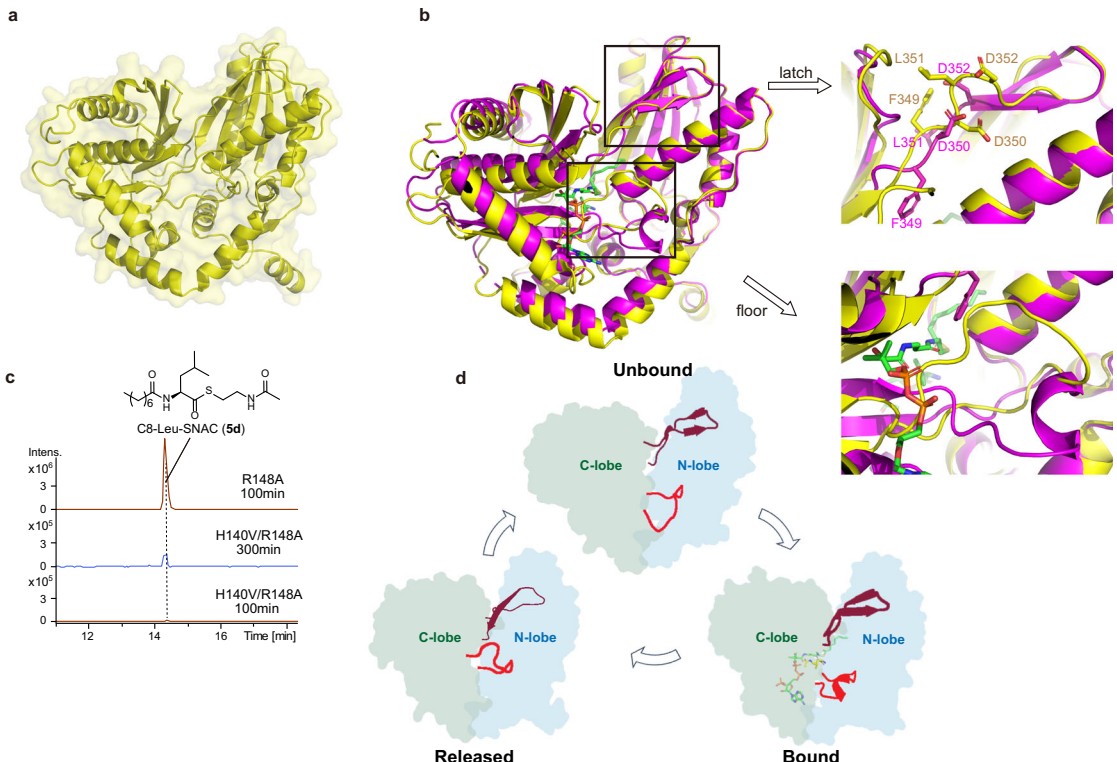

**Fig. 9 Potential conformational changes within the reaction cycle of lipoinitiation. a** The overall structure of RzmA-Cs H140V/R148A in a "product-released" state. **b** Superimposition of H140V/R148A structures in "released" (yellow) and "bound" (magenta) forms. Two enlarged views highlighting the changes in the "latch" and "floor" regions. **c** The comparative analysis of C8-Leu-SNAC (**5d**) in vitro produced by RzmA-Cs mutants R148A and H140V/R148A with substrates C8-CoA and L-Leu-SNAC. EICs at $m/z = 359.2$ (**5d**, $[M + H]^+$). **d** A schematic diagram showing the potential conformational changes of Cs domains during the reaction cycle, with the "latch" and "floor" regions (secondary structures in cartoon mode) highlighted in brown and red, respectively.

substrates shown in vitro (Fig. 3c, d) support full-length Cs domain swapping, but the specificity of the contiguous A domain should also be considered to maximize the production of engineered products with different acyls. The suitable recombination sites between the Cs and A domains need to be further investigated in detail for the efficient engineering of lipopeptides. The recently reported research on the modification of NRPSs by A domain alone substitution also supported that C domain specificities were not as strict as we previously thought[36]. Our results suggested that the use of full-length Cs domain as a swapping unit is a feasible way to modify lipopeptides.

On the basis of the complex structure of the Cs domain with acyl-CoA and of both in vitro and in vivo mutation experiments in this investigation, we revealed the binding pocket of acyl group and the key amino acid residues that control the length of the acyl group. A recent study using modeling and in vitro experiments identified four residues as functionally related to the fatty acyl substrate selectivity of LptA-C1[12], but the residues were different from our revealed key sites demonstrated by the cocrystal structure and by in vitro and in vivo experiments. We found that three sites in RzmA-Cs Q36, Y138, and R148 play key roles in controlling the specificity of the acyl chain, in contrast to the previous study on LptA-C1 in A54145 biosynthetic pathway[12], i.e., A152, A369, A386, and L397 (corresponding to Y149, D350, N367, and Y378 in RzmA-Cs) (Supplementary Fig. 4a). We also noticed the inconsistencies between our in vivo and in vitro experiments, due to the complexity of the in vivo system and/or to different products. The in vitro experiments determined only the formation of the first simple biosynthetic intermediate mimics by a Cs domain-catalyzed reaction, while the in vivo experiments

showed the final complex products after many reactions. The concentrations of the acyl-CoA substrates vary widely in cells; e.g., acetyl-CoA, as an important primary metabolite, is present at concentrations at least one or two orders of magnitude higher than those of other medium or long fatty acyl-CoAs in *E. coli*[44]. We set up another in vitro experiment to simulate in vivo conditions using higher concentrations of acetyl-CoA and butanoyl-CoA (C2-CoA:C4-CoA:C6 to C18-CoA = 1000:100:1) and an extended reaction time, and the products C2-Leu-SNAC and C4-Leu-SNAC were detected with relatively high yields, as shown by the counterparts in the in vivo experiments (Supplementary Fig. 10). This result signified that substrate concentration and reaction time also influenced the in vitro reaction, suggesting that the donor specificity of the Cs domain was not very strict, although the optimal products could be obtained. Triple mutations in the RzmA-Cs domain (Y138A/M143A/R148G) can greatly extend the acyl chain of rhizomide A from C2 to C16 in vivo and in vitro, which proved that point mutations in the Cs domain can change the acyl chains of lipopeptides. Thus, we may establish a foundation to artificially generate lipopeptides with varying lengths of acyl chains for biological evaluation. However, more combinatorial mutations of the key sites in the acyl-CoA building pocket are required to achieve the precise control of acyl chain length. This Cs domain bioengineering strategy including domain swapping and point mutation, is a complement to attempts to modify the peptidyl backbone in the field of combinatorial biosynthesis to produce novel lipopeptides and non-ribosomal peptides.

The co-complex structures of an enzyme and its substrates or intermediate analogs usually provide direct evidence for the

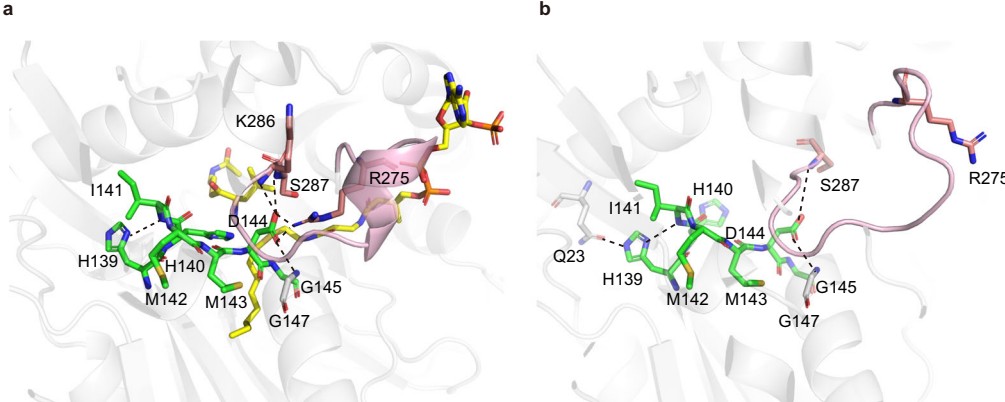

**Fig. 10 Potential structure-stabilizing function of the conserved "HHxxxDG" motif. a** As revealed by the co-complex structure of RzmA-Cs with two substrates, the side-chain of H139 interacts with the main chain of I141, and that of D144 interacts with the side-chain of R275 (to form a salt bridge) and the main chain atoms of G147, K286, and S287. **b** In the "unbound" structure of RzmA-Cs, most of the above residues interacting with H139 and D144 are at the similar positions, while the R275 residue of the "floor" region is far away from D144. The seven residues in the conserved "HHxxxDG" motif are colored in green, and the "floor" region is highlighted in light pink. The two bound substrates are colored yellow. All interactions are shown as black dashed lines.

catalytic mechanism. For the NRPS Cs domains, how the conserved "HHxxxDG" motif participates in condensation reactions is an interesting question that is not yet fully answered. A previous study on the CdaPS1-Cs domain suggested that the "catalytic histidine" (second histidine within the motif) is responsible for positioning the α-amino group of acceptor substrates[19]. Our co-complex structures of RzmA-Cs not only supported this conception, but also extended the positioning function of this histidine (H140) to the donor substrates, i.e., acyl-CoAs. In addition, we infer that the glycine within the motif (G145) is conserved as a residue without a side-chain to make room for the donor substrates. The other two conserved residues of the motif (H139 and D144 in RzmA-Cs), may help maintain the appropriate architecture around the active site, since their side chains are involved in multiple interactions with neighboring residues (Fig. 10a). It is noteworthy that among these D144-interacting resides, R275 undergoes a very large conformational change from the "unbound" status to contact D144 (Fig. 10b), as does the "floor" region of RzmA-Cs (see below). This salt bridge between D144 and R275 is potentially critical for the proper structure and function of Cs domains, as both residues are highly conserved in the sequence alignment (Supplementary Fig. 4a).

NRPSs are megaenzymes that conduct peptide biosynthesis like an assembly line, which requires cooperative conformational changes in all domains and modules. Recent structural studies on multidomain NRPS proteins visualized the conformational flexibility among different domains even modules during the biosynthetic reaction[21–24,27]. However, other than the various degrees of openness between the two lobes of the C domains[39], the conformational changes in this domain during the condensation process have not been fully revealed, due to the lack of co-complex structures with substrates. Fortunately, we were able to capture several snap-shots of the conformations of RzmA-Cs, illustrating a potential reaction cycle including the "unbound-bound-released" states (Supplementary Video 1). The most dramatic changes occurred in the "latch" and "floor" regions, which together define the shape and size of the active-site tunnel. In particular, the position of the "floor" loop in the "released" state overlapped with the binding position of CoA in RzmA-Cs, suggesting that this loop may help push the CoA out of the donor binding site after condensation (Fig. 9b). This function of the "floor" loop echoes previous findings that its counterpart loop in the epimerization domain of gramicidin S synthetase (GrsA),

interacts directly with the neighboring PCP domain at the donor site[45]. Further investigations are much warranted to determine how the conformational changes of C domains function in concert with larger domain movements within a module or even across modules of NRPSs.

Collectively, here, our genetic engineering and cocrystal structures revealed how Cs domains recognize and interact with both donor and acceptor substrates, and we observed their unique conformational changes during the reaction cycle. Our work provides further insights into the mechanism of lipoinitiation, and suggests a feasible approach to generate lipopeptides with various acyl chains, facilitating future bioengineering efforts to produce more novel nonribosomal peptides.

## Methods

**General methods**. All bacteria strains, plasmids, and sequences of primers used in this study are supplied in Supplementary Data 1. Detailed description for the construction of plasmids and related strains/plasmids/mutants are also given in Supplementary Data 1.

The *Paraburkholderia rhizoxinica* HKI 454 (DSM 19002), *Schlegelella brevitalea* DSM 7029 and its mutants were cultured in CYMG (8 g/l Casein peptone, 4 g/l Yeast extract, 4.06 g/l MgCl₂·2H₂O, 10 ml/l glycerin) broth or agar plates with apramycin (20 µg/ml), kanamycin (20 µg/ml), gentamicin (15 µg/ml), or chloramphenicol (15 µg/ml) as required. The *E. coli* cells for recombineering and plasmid propagation were cultured in Luria-Bertani (LB) broth or on LB-agar plates (1.2% agar) with ampicillin (100 µg/ml), kanamycin (15 µg/ml), chloramphenicol (15 µg/ml), or gentamicin (5 µg/ml) as required. Reagents were purchased from New England Biolabs, Thermo Fisher Scientific, Invitrogen, and Sigma-Aldrich. DNA primers were purchased from Tsingke Biological Technology and Sangon Biotech.

*Cloning and heterologous expression of rhizomide and holrhizin biosynthetic gene clusters.* The rhizomide and holrhizin gene clusters, *rzmA* (RBRH_RS12370) and *holA* (RBRH_RS16800), were directly cloned from the genomic DNA (Genbank: NC_014718.1) of *P. rhizoxinica* HKI 454 using Red/ET recombineering in *E. coli*, respectively. The direct cloning experiments were performed according to the previous protocols[46–48]. In brief, genomic DNA is digested by appropriated restriction enzymes and purified by ethanol precipitation. The digested genomic DNA, p15A vector with homology, T4 polymerase, and buffer are mixed and reacted in a thermocycler in program: 25 °C, 60 min; 75 °C, 20 min; 50 °C, 60 min. The mixture is then electroporated into the L-arabinose induced *E. coli* GB05-dir (Supplementary Data 1—Bacteria strains) competent cells and plated on appropriated LB-agar plates to incubate at 37 °C overnight. The *rzmA* (23.8 kb) and *holA* (22.3 kb) regions were cloned from digested genome mixture (ScaI and SalI for *rzmA*, EcoRV for *holA*) into the corresponding p15A-cm vectors in *E. coli*, and the resulting plasmids p15A-cm-rzmA and p15A-cm-holA were then inserted transferring *IR-oriT-tnpA-IR-km* cassettes to get final expression plasmids p15A-tnpA-km-rzmA and p15A-tnpA-km-holA, respectively. The *rzmA* and *holA* gene clusters

are under the control of $P_{Tn5-km}$ constitutive promoter[33]. The primers for direct cloning were listed in Supplementary Data 1.

Then, both plasmids were electroporated into the heterologous host *Schlegelella brevitalea* DSM 7029 (previous known as Burkholderiales strain DSM 7029 or *Polyangium brachysporum* K481-B101)[33,34,49] for functional expression according to the previous method[29,33]. The correct colonies were checked by colony PCR. The primers were listed in Supplementary Data 1. The verified mutants containing *rzmA* or *holA* were cultured overnight and inoculated into 50 ml antibiotic supplemented CYMG medium in a 250 ml flask and incubated at 30 °C, 200 rpm for 4 days. The resin Amberlite XAD-16 (2%) was added into the culture and continually incubated for another day. The cells and resins were collected by centrifugation and extracted with 40 ml MeOH for 2 h. The supernatant was concentrated in vacuo and the resulting residue was dissolved in 1 ml MeOH for further metabolic analysis by UPLC-MS or UPLC–HRMS.

*Cs (sub)domain swapping.* The Cs domain swapping was also conducted by LLHR of recombineering[46]. The procedure was as same as did in direct cloning experiments by using linear DNA and digested vectors. For *rzmA* gene cluster, a linear fragment harboring HolA-Cs domain (NCBI: WP_041754829.1, residues 1–479) and *km-IR-tnpA-oriT-IR* cassette flanked with homology arms were amplified from p15A-tnpA-km-holA by PCR. The p15A-cm-rzmA was linearized by digestion of ScaI and NheI. Two linear fragments were recombined to form plasmid p15A-tnpA-km-rzmACsholA (RCH in Fig. 2a, c) in which the RzmA-Cs domain (NCBI: WP_013428324.1, residues 1–479) was replaced by HolA-Cs domain (Fig. 2a). Likewise, plasmid p15A-tnpA-km-holACsrzmA (HCR in Fig. 2d) was also constructed where the HolA-Cs was changed to RzmA-Cs domain, except the p15A-cm-holA was linearized by digestion of HindIII and EcoRV. The swapping of Cs subdomain (N-lobe of Cs domain) followed the same procedure as the full-length Cs swapping method by using fragments RzmA-CsN (residues 1–188) or HolA-CsN (residues 1–189) and corresponding original plasmids p15A-cm-rzmA and p15A-cm-holA were linearized by digestion of ScaI/BspQI and EcoRV (Fig. 2b–d), respectively. The resulting plasmids were verified and transformed into *S. brevitalea* DSM 7029 for heterologous expression and metabolic analysis. Detailed description for the construction of plasmids and related strains/mutants/primers were given in Supplementary Data 1.

For glidobactin gene cluster (*glb*), the Cs domain swapping was performed in the native producer *S. brevitalea* DSM 7029 using Redαβ7029 recombineering[29]. HolA-Cs and GlpC-Cs domains were used to replace GlbF-Cs domain. First, the GlpC-Cs encoding region in DSM 7029 was replaced by a gentamicin resistance gene to obtain mutant 7029ΔGP avoid interference of homologous recombination. Then an *apra-phiC31* cassette was used to replace the *tnpA* in p15A-tnpA-km-holA or p15A-tnpA-km-rzmA to form the cassette, respectively. Then this cassette flanked with homology arms was amplified by PCR and it was to replace the GlbF-Cs domain in 7029ΔGP. The resulting mutant 7029HΔGP or 7029RΔGP contains a chimeric *glbF* with HolA-Cs or RzmA-Cs under the control of $P_{Tn5-km}$ promoter. The *km-apra* cassette was connected to GlpC-Cs encoding region by overlapping PCR, then it was used to replace the GlbF-Cs domain in 7029ΔGP to get mutant 7029PΔGP. N-terminal GlpC-Cs hybrid *glbF* was constructed in the same way by using the overlapped GlpC-CsN-*km-apra* cassette (7029PNΔGP). All mutants were verified by colony PCR and sequencing. The correct mutants were cultured and analyzed as mentioned above. The $P_{Tn5-km}$ promotor was also inserted in front of *glbF* to make a control for metabolic analysis (7029BΔGP). Detailed description for the construction of *amp-ccdB* plasmids and related strains/plasmids/mutants were given in Supplementary Data 1.

*Point mutations of Cs domains in* rzmA *and* holA *gene clusters.* The seamless point mutations in the large biosynthetic genes were created by CcdB counterselection-based recombineering reported previously[50,51]. This method mentioned two rounds of recombineering: the site of interest was first replaced by a counter-selection cassette *amp-ccdB* in *E. coli* GBred-gyrA462; then a synthetic single-strand oligonucleotide containing a mutation of interest to replace this cassette under the counterselection of the toxicity marker inferred by the counter-selectable gene *ccdB*. All single mutation R148M, R148L, R148V, R148A, R148G of RzmA-Cs in *rzmA* (p15A-tnpA-km-rzmA and pET28a-rzmACs) and A149M, A149L, A149V, A149R, A149G of HolA-Cs in *holA* (p15A-tnpA-km-holA and pET28a-holACs) were introduced using this method and verified by sequencing, respectively. The double or multiple mutants including Q36G/R148G (Q36G), Q136A/R148G (Q136A), Y138A/R148G (Y138A), M143A/R148G (M143A), Q136A/Y138A/R148G (QYA), Y138A/M143A/R148G (YMA), and Q136A/Y138A/M143A/R148G (QYMA), H140V/R148A of RzmA-Cs in *rzmA* (p15A-tnpA-km-rzmA and pET28a-rzmACs) were also achieved using repeated counterselection recombineering. All primers used for point mutations listed in Supplementary Data 1. The correct plasmids containing biosynthetic gene clusters were transferred into *S. brevitalea* DSM 7029 for heterologous expression and metabolic analysis, respectively. The Cs domain variants were transferred into *E. coli* BL21 for protein expression and purification. Full information for the construction of plasmids and related strains/mutants/primers were given in Supplementary Data 1.

*Metabolic analysis.* The metabolic analysis of mutants was performed by UPLC-MS and UPLC–HRMS. The UPLC-MS was same as mentioned in our previous

publication[52]. The UPLC–HRMS analysis was same as described before except the UPLC system which was performed on a column (YMC, $C_{18}$, 250 × 4.6 mm, 5 μm) with gradient elution at a constant flow rate of 1 ml/min, 0–5 min, 5% B; 5–55 min, 5–95% B; 55–60 min, 95% B[29]. Mass spectra were acquired in *m/z* in a positive ionization mode with auto MS2 fragmentations. All small-scale fermentations and analyses followed the same method as described above.

*Purification of compounds.* The derivatives of rhizomide A and holrhizin A were purified according to our previous description with minor modifications[29,30]. For compound **1d**, the extract of 12 L culture of mutant 7029WT::p15A-tnpA-km-rzmACsholA (7029RCH, Fig. 2c) was eluted by a normal-phase silica gel column resulting in fraction of $CH_2Cl_2$:MeOH = 15:1 which was further purified by semipreparative reversed-phase high-performance liquid chromatography (RP-HPLC) (column: YMC-Pack ODS-A $C_{18}$; 5 μm, 250 × 10 mm; gradient elution: 0–5 min, 65% ACN; 5–25 min, 65–68% ACN; 25.1–35 min, 95% ACN) to afford **1d** (6.4 mg, white amorphous solid) at retention time 21.5 min. For compound **2a**, the extract of 12 L culture of mutant 7029WT::p15A-tnpA-km-holACsrzmA (7029HCR, Fig. 2d) was eluted by gradient elution resulting in fraction of $CH_2Cl_2$:MeOH = 10:1 which was further purified by semipreparative RP-HPLC (column: YMC-Pack ODS-A $C_{18}$; 5 μm, 250 × 10 mm; gradient elution: 0–5 min, 30% ACN; 5–25 min, 30–45% ACN; 25.1–35 min, 95% ACN) to obtain **2a** (5 mg, white amorphous solid) at retention time 21.6 min. The **1d** and **2a** were dissolved into DSMO-$d_6$ for NMR recording (Bruker Avance III-600 spectrometer) and the NMR data were summarized in Supplementary Tables 1–3 and the NMR spectra were shown in Supplementary Figs. 11–20, respectively. For compounds **1e–1h**, the extract of 20 L culture of mutant 7029WT::p15A-tnpA-km-rzmA-R148G-Y138A-M143A (7029YMA, Fig. 7a) was eluted by a normal-phase silica gel column. The elution of $CH_2Cl_2$:MeOH = 20:1 was further purified by HPLC (YMC-Pack ODS-A $C_{18}$; 5 μm, 250 × 10 mm, gradient elution 0–3 min, 74% ACN; 3–35 min, 74–100% ACN; 35.1–42 min 100% ACN) to yield **1e** (7 mg) at 12.5 min, **1f** (12 mg) at 18.5 min, **1g** (12 mg) at 26.5 min, and **1h** (12 mg) at 36 min, respectively. The NMR data were summarized in Supplementary Tables 1, 2, and the NMR spectra were shown in Supplementary Figs. 21–28. For compounds **7b** and **7c** (Fig. 3a, b), the extract of 12 L culture of mutants Redαβ7029glpCCs::genta glbFCs::$P_{Tn5-km}$-apra-holACs (7029HΔGP for compound **7b**) and Redαβ7029glpCCs::genta glbFCs::$P_{Tn5-km}$-apra-glpCCs (7029PΔGP for compound **7c**), were eluted by a normal-phase silica gel column, respectively. For compound **7b**, gradient elution resulting in 9 fraction of $CH_2Cl_2$:MeOH = 8:1 which was further purified by HPLC (YMC-Pack ODS-A $C_{18}$; 5 μm, 250 × 10 mm, gradient elution 0–5 min, 30% ACN; 5–25 min, 30–55% ACN; 25.1–35 min 95% ACN) to yield compound **7b** (11 mg) at 21 min. For compound **7c**, gradient elution resulting in 9 fraction of $CH_2Cl_2$:MeOH = 8:1 which was further purified by HPLC (YMC-Pack ODS-A $C_{18}$; 5 μm, 250 × 10 mm, gradient elution 0–5 min, 35% ACN; 5–25 min, 35–55% ACN; 25.1–35 min 95% ACN) to yield compound **7c** (11 mg) at 21 min. The NMR data of **7b** and **7c** were summarized in Supplementary Table 4 and the NMR spectra were shown in Supplementary Figs. 29–32.

*Compound characterization.* **1d**, **1e**, **1f**, **1g**, **1h**, **2a**, **7b**, and **7c**: $^1H$ NMR and $^{13}C$ NMR were summarized in Supplementary Tables 1–4; For **1d**, UV: $\lambda_{max}$ 224 nm; HRMS (*m/z*): $[M + H]^+$ calcd. for $C_{41}H_{65}N_7O_{10}$, 816.4826; found, 816.4832; For **1e**, UV: $\lambda_{max}$ 227 nm; HRMS (*m/z*): $[M + H]^+$ calcd. for $C_{43}H_{69}N_7O_{10}$, 844.5139; found, 844.5132; For **1f**, UV: $\lambda_{max}$ 228 nm; HRMS (*m/z*): $[M + H]^+$ calcd. for $C_{45}H_{73}N_7O_{10}$, 872.5452; found, 872.5429; For **1g**, UV: $\lambda_{max}$ 228 nm; HRMS (*m/z*): $[M + H]^+$ calcd. for $C_{47}H_{77}N_7O_{10}$, 900.5765; found, 900.5741; For **1h**, UV: $\lambda_{max}$ 228 nm; HRMS (*m/z*): $[M + H]^+$ calcd. for $C_{49}H_{81}N_7O_{10}$, 928.6078; found, 928.6070; For **2a**, UV: $\lambda_{max}$ 223 nm; HRMS (*m/z*): $[M + H]^+$ calcd. for $C_{36}H_{56}N_6O_{10}$, 733.4091; found, 733.4117; For **7b**, UV: $\lambda_{max}$ 235 nm; HRMS (*m/z*): $[M + H]^+$ calcd. for $C_{23}H_{40}N_4O_6$, 469.2981; found, 469.3007; For **7c**, UV: $\lambda_{max}$ 235 nm; HRMS (*m/z*): $[M + H]^+$ calcd. for $C_{25}H_{44}N_4O_6$, 497.3294; found, 497.3304.

*Bioinformatic analysis.* The sequence alignment of Cs domains was conducted by ClustalW2 and then processed by ESPript 3.0[53,54]. The structures of Cs domains were predicted by SWISS-MODEL with tunnels calculated by Mole 2.0[55,56].

*Synthesis and purification of aminoacyl-SNACs.* Aminoacyl-N-acetylcysteamine thioesters (SNACs) were synthesized as imitative substrates by the following method reported previously[57]. L-Leu-SNAC (**4**) and L-Val-SNAC (**6**) were purified by semipreparative RP-HPLC (column: YMC-Pack ODS-A $C_{18}$; 5 μm, 250 × 10 mm, gradient elution: 0–5 min, 2% MeOH; 5–25 min, 2–30% MeOH; 25.1–35 min, 95% MeOH; 35–40 min, 2% MeOH) to yield L-Leu-SNAC (**4**, 180 mg) at retention time ~15.2 min and L-Val-SNAC (**6**, 230 mg) at retention time ~15.0 min, respectively. L-Ala-SNAC and L-Met-SNAC were purified by gradient elution (0–5 min, 5% MeOH; 5–25 min, 5–35% MeOH; 25.1–35 min 95% MeOH) to yield L-Ala-SNAC at ~10.5 min and L-Met-SNAC at ~23.5 min, respectively. L-Phe-SNAC was purified by gradient elution (0–5 min, 25% MeOH; 5–25 min, 25–40% MeOH; 25.1–35 min, 95% MeOH) to yield at ~17.8 min. L-Ser-SNAC was purified by gradient elution (0–4 min, 5% MeOH; 4–22 min, 5–16% MeOH; 22.1–30 min, 100% MeOH) at ~13 min. L-Thr-SNAC was purified by gradient elution (0–4 min, 8% MeOH; 4–22 min, 8–16% MeOH; 22.1–30 min, 100% MeOH) at ~11 min.

*Protein expression and purification.* RzmA-Cs (NCBI: WP_013428324.1, residues 1–431), HolA-Cs (NCBI: WP_041754829.1, residues 1–431), and GlbF-Cs (NCBI: AKJ29077.1, residues 1–458) encoding regions were amplified by PCR and cloned into pET28a (+) vector by LLHR of recombineering[46]. The correct plasmids were electroporated into *E. coli* BL21 (DE3) for protein expression. The 40 ml overnight culture was inoculated into 2 L LB medium supplemented with 5 μg/ml kanamycin and cultivated for 2.5 h at 37 °C, 200 rpm. Then the bacteria were induced overnight with 0.1 mM isopropyl-1-thio-β-D-galactopyranoside (IPTG) after precooled for 3 h at 18 °C, 200 rpm. *E. coli* cells expressing RzmA-Cs, HolA-Cs, and GlbF-Cs were harvested and resuspended into 80 ml wash buffer (50 mM Tris-HCl, 300 mM NaCl, and 50 mM imidazole, pH = 7.5), and further lysed by sonication, respectively. Debris was removed by centrifugation for 1 h at 23,447 × *g* before the supernatant being applied onto Ni-NTA agarose (Bestchrom). N-terminal hexa-histidine-tagged proteins were digested with 140 U thrombin enzyme (Solarbio) for 14 h at 4 °C after washed out impurity proteins as possible with 200 ml wash buffer and balanced with analysis buffer (50 mM Tris-HCl, 300 mM NaCl, pH = 7.5). Tag-removed proteins were collected and concentrated in Amicon® Ultra-15 30 K device (Millipore) before applied into size-exclusion chromatography by using the ÄKTA prime PLUS system (GE) mount with Superdex 200 pg 16/600 column. All proteins were purified in similar method.

*Enzymatic activity assay.* The purified Cs domain and mutants were subjected to enzyme activity assay. The substrates including aminoacyl-SNACs were stocked at −80 °C in 3% trifluoroacetic acid, the acyl-CoAs including acetyl-CoA (C2-CoA), butanoyl-CoA (C4-CoA), hexanoyl-CoA (C6-CoA), octanoyl-CoA (C8-CoA), decanoyl-CoA (C10-CoA), dodecanoyl-CoA (C12-CoA), myristoyl CoA (C14-CoA), palmitoyl-CoA (C16-CoA), and stearoyl-CoA (C18-CoA) were purchased from Sigma-Aldrich. For single assay, 20 μM protein was mixed with 8 mM aminoacyl-SNAC and 80 μM acyl-CoA in the reaction buffer (50 mM Tris-HCl, 300 mM NaCl, 1 mM DTT and 20% glycerol, pH = 7.5) to final volume of 20 μl. The mixture was incubated at 30 °C for 100 min, and reaction was quenched with 20 μL MeOH. The resulting mixture was centrifuged to remove the protein and 2 μl of supernatant was used for UPLC-MS analysis. For the substrate competition assay, we used the mixture of 80 μM of each acyl-CoA as donor substrates to investigate the specificities of RzmA-Cs and its variants for acyl-CoAs. The relative yield of each product was determined by comparison of its peak area in the UPLC-MS chromatogram. The H140V/R148A mutant was tested by similar procedure in a series of time points, 0 min, 100 min, 300 min, 10 h, and 24 h, with modified substrates and protein concentrations (100 mM protein, 80 mM L-Leu-SNAC, and 2 mM C8-CoA). All tests were conducted in triplicates.

For the competition assay which simulates in vivo conditions, longer incubation time and different portions of acyl-CoAs were used. The 1.5 mM acetyl-CoA plus 150 μM butanoyl-CoA and 1.5 μM others acyl-CoAs (from C6-CoA to C18-CoA) were mixed with 8 mM L-Leu-SNAC and 20 μM protein in reaction buffer for incubated 6 h at 30 °C as described above. Extraction and analysis methods were same as described above. All tests were conducted with three replicates.

Four Cs domains were chosen to tested specificities of acceptor substrates, including RzmA-Cs, HolA-Cs, and GlbF-Cs. Seven different L-aminoacyl-SNACs including L-Val-SNAC, L-Leu-SNAC, L-Ala-SNAC, L-Phe-SNAC, L-Met-SNAC, L-Ser-SNAC, and L-Thr-SNAC were used as mimic acceptor substrates, and were checked with respective donor substrates, i.e., RzmA-Cs was tested with acetyl-CoA; HolA-Cs was tested with octanoyl-CoA and GlbF-Cs was checked with myristoyl-CoA by following procedure[14]. Twenty micromolar of proteins were mixed with 5 mM aminoacyl-SNAC and 160 μM acyl-CoA in the reaction buffer as described above. After incubation at 30 °C for 120 min, mixtures were extracted and analyzed as mentioned above. The relative yield of each product was determined by comparison of its peak area in the UPLC-MS chromatogram and the native product of mimic substrates were used as references, i.e., as the 100%. All tests were conducted in triplicates.

*Protein thermal shift assay.* The purified protein was dissolved in analyze buffer (50 mM Tris-HCl, 300 mM NaCl, pH = 7.5) to reach the final concentration 4 μM, and tested by Protein Thermal Shift Dye Kit (ABI Lot: 00710602). The reaction mixture (20 μl) was operated in Applied Biosystems Quant Studio™ 3 Real-Time PCR System (Applied Biosystems Inc., CA, USA) in the condition as follows: step 1: 25 °C, 1.6 °C/s and step 2: 99 °C, 0.05 °C/s in the setting of excitation filter: x4 (580 ± 10 nm) and emission filter: m4 (623 ± 14 nm). Melting curve data were analyzed by Protein Thermal Shift Software V1.30 (Applied Biosystems Inc., CA, USA) and ΔTm (Boltzmann equation) was calculated.

*Isothermal titration calorimetry.* The purified RzmA-Cs R148A protein was dialyzed in analyze buffer (50 mM Tris-HCl, 100 mM NaCl, pH = 7.5) overnight. C8-CoA was dissolved in a dialysis buffer. The binding affinity between C8-CoA and R148A was determined using ITC on PEAQ-ITC (Malvern). C8-CoA at 400 μM was titrated into 400 μM of R148A in the cell in dialysis buffer consisting of 50 mM Tris-HCl, 100 mM NaCl, pH = 7.5.

*Crystallization and X-ray data collection.* Crystallization of RzmA-Cs WT, R148A, and H140V/R148A proteins in the unbound form was carried out using sitting drop vapor diffusion method at 16 °C, by mixing equal volume of protein

(10 mg/ml) and the reservoir containing 30 mM citric acid, 70 mM Bis-Tris propane (pH 7.6), and 20–25% PEG 3350. The crystals of all co-complex and the "product-released" form were grown in the same precipitant solution of 100 mM Bis-Tris propane (pH 9.0) and 30–35% PEG 6000 using hanging drop vapor diffusion method at 16 °C. Crystals of binary complex were obtained by R148A and H140V/R148A (10 mg/ml), respectively, co-crystallized with 2 mM C8-CoA. Crystal of ternary complex was obtained by H140V/R148A (10 mg/ml) co-crystallized with 2 mM C8-CoA and 5 mM L-Leu-SNAC in 3 days, and ligand-free crystals in "product-released" status were obtained at the same drop in 2 weeks. Crystals were flash-frozen directly from the growing drops without additional cryo-protectants. Diffraction data of R148A + C8-CoA was collected at 100 K on beamlines BL17U1 at the Shanghai Synchrotron Radiation Facility (SSRF)[58], and processed using XDS package[59]. The other diffraction data were collected on beamlines BL19U1 at the SSRF[60]. The data of RzmA-Cs WT, R148A, H140V/R148A, and H140V/R148A + C8-CoA+Leu-SNAC were processed using the HKL2000 program[61]. The data of H140V/R148A + C8-CoA and H140V/R148A product-released were processed using the HKL3000 program[62].

*Structure determination and refinement.* The structure of RzmA-Cs WT was solved by molecular replacement with the program Phaser[63], using the CdaPS1-Cs (PDB code: 4jn3) structure as the search model[18]. Further manual model building was facilitated by using Coot[64], combined with the structure refinement using Phenix[65]. Likewise, the other structures were solved by molecular replacement using RzmA-Cs WT as the search model and further refined. Data collection and structure refinement statistics are summarized in Supplementary Table 5. The Ramachandran statistics, as calculated by Molprobity[66], are 97%/0.4%, 95%/0.4%, 97%/0%, 96%/0.3%, 99%/0%, 97%/0.2%, and 99%/0.2% (favored/outliers) for structures of RzmA-Cs WT, R148A, R148A + C8-CoA, H140V/R148A + C8-CoA, H140V/R148A + C8-CoA+Leu-SNAC, and H140V/R148A product-released, respectively. All the structural figures and the video were prepared using PyMol (The PyMol Molecular Graphics System, Version 2.3, Schrödinger, LLC).

**Reporting summary.** Further information on research design is available in the Nature Research Reporting Summary linked to this article.

## Data availability
The data that support the findings of this study are available within the paper and its Supplementary Information files. A reporting summary for this article is available as a Supplementary Information file. The bacteria materials generated during the current study are available from the corresponding author upon reasonable request. Coordinates and structure data that support the findings of this study have been deposited in the Protein Data Bank with the accession codes 7C1H, 7C1K, 7C1L, 7C1P, 7C1R, 7C1S, and 7C1U (Supplementary Table 5). Source data are provided with this paper.

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

## Acknowledgements

We would like to thank Zhifeng Li, Jingyao Qu, Xiaoju Li, and Haiyan Sui from State Key Laboratory of Microbial Technology of SDU for data collection of UPLC–HRMS, NMR,

and X-ray diffraction, and Dr. V. Ravichandran (SDU) for proofreading of the manuscript. We also thank the staff from beamlines BL19U1 and BL17U1 at Shanghai Synchrotron Radiation Facility (SSRF) for assistance during data collection. This study was supported by the National Key R&D Program of China (2019YFA0905700, 2017YFD0201400, 2018YFE0113000), the National Natural Science Foundation of China (31670098, 32070060, 31670097, 31700114), the Shandong Provincial Natural Science Foundation, China (ZR2019JQ11), the Natural Science Foundation of Jiangsu Province (BK20170399), the 111 project (B16030), the Taishan Scholars Program of Shandong Province (tsqn201909004), and the Qilu Young Scholar Startup Funding and Youth Interdisciplinary Innovative Research Group (2020QNQT009) of SDU.

## Author contributions

X.B. and D.W. conceived the study; X.B., D.W., and Y.Z. supervised the experiments; L.Z., N.Z., H.C., H.Z., X.B, Y.Z., and X.B. constructed mutants and expressed plasmids, analyzed metabolic data and purified compounds, isolated the proteins, and conducted biochemical studies; L.Z., X.D., and N.Z. carried out crystallizations; X.D., F.L., X.R., and D.W. collected X-ray diffraction data, solved, and refined the structures; L.Z., X.D., D.W., and X.B. wrote the manuscript with the input from all authors.

## Competing interests

The authors declare no competing interests.
