## [Peer Review File · Nature Communications]

REVIEWER COMMENTS

Reviewer #1 (Remarks to the Author):

Overview

The structures and rational mutagenesis studies presented in the manuscript submitted by Lin Zhong and colleagues are really very interesting, and likely to hold substantial interest for others in the field. However, there are some inconsistencies between some of the experiments that need to be discussed in greater detail. Some of the conclusions are also a bit too broad, given the focus on just two enzymes, and would require further experimental data to substantiate them. Additional explanations are needed to clarify some of the work, and there is a strong need for a professional proof-reading service – nearly every sentence contains typographical errors and/or grammatical incongruities, and in some places these greatly obscure the intended meaning.

Specific comments

Lines 87 to 89. Raw numbers of percentage identity are not overly informative. An alignment of these two Cs domains would be helpful to have in the supplementary materials and could also be used to exemplify the recombination points used for Cs domain and subdomain substitution (see next point).

Lines 89 to 92. Very little information was provided as to the rationale for selecting subdomain recombination points. I request that the location be noted precisely in the alignment suggested above – this would greatly aid the reader.

Lines 97 to 99. Given the sample size of two, and that only one recombination point was tested, there is very little evidence to conclude that Cs domain substitution is more reliable than subdomain substitution. More substitutions employing Cs domains from a range of different NRPSs and several different recombination points would be needed to make such a statement with any confidence.

Lines 100 to 102. It is already known that the Cs domain can define specificity of lipid chains - indeed, the authors specifically mention that this is known in lines 42 to 46. However, this sentence now makes it sound as if the authors are claiming that they have observed this as a new discovery. In addition, ref 12 cited in this manuscript already shows that it is feasible to alter Cs domain donor site specificity with point mutations. Thus, the authors need to tone down their claims and put them in context so that it does not sound like they are seeking credit for discovering previously-known phenomena.

Lines 116 to 119. Figures 3c and S1e do not really seem to be in agreement, as the authors claim here. Figure 3c shows the major products for the R148G variant to contain a lipid with $n=0$ and $n=6$, whereas the major products for the R148G variant in Figure S1e are predominantly lipids with $n=6$ and $n=8$. Was R148G tested with all of 5a to 5e, or only 5d/5e? Or is there some greater inconsistency between the in vivo experiments and in vitro assay? A more in-depth discussion is required here, and the legend to Figure S1 requires substantially more detail, to explain exactly what was tested, and how.

Figures 5e and 5f. Again, there appear to be inconsistencies between the in vivo and in vitro data that the authors do not acknowledge. For example, the major product in vivo is frequently the C2-Rzm, but this is not observed at all for most of the variants in vitro. Were all substrates tested in all cases? The graphs appear to be drawn in R and the positioning of the bars suggests that either not all substrates were tested, or some data has been omitted. As per my previous comment, the legend requires substantially more detail to clearly explain how each set of experiments was performed. Please also note that the choice of colour scheme makes it very difficult to judge differences between C2-Leu-SNAC and C8-Leu-SNAC in 5f.

Figures 5e and 5f. Another inconsistency that is sufficiently prominent I have noted it as a separate point - why is it that yield of the C2 product was so low for the R148 WT in Figure 5f, when it was one of the highest yields in vivo? Conversely, why are mutants that had very low yield in Figure 5e suddenly high yielding in Figure 5f?

Figure 3c and 5e: R148G is analysed in vivo in both Figures 3c and 5e. In 3c the major products are $n = 0$ and $n = 6$, with no trace of $n = 8$; however, in 5e the major products have changed to be $n = 0$ and $n = 8$, with no trace of $n = 6$. It is unclear what difference there is between these experiments.

Lines 160 to 163: Does mere proximity suggest the histidine is involved in substrate positioning? Is the substrate orientation consistent with this hypothesis and is there any hydrogen bonding supporting the positioning role? These assumptions do receive some later experimental support when this histidine is substituted to a valine (lines 196-200), but at this earlier stage of the text, providing additional guidance to the reader would be useful.

Lines 225 to 235: The authors identify putative substrate-determining residues and make the argument that these might be relevant for C domains with different acceptor substrates. However, Figure S4a does not explain which substrate each C domain accepts, which would be useful information. Indeed, to support their argument, it would be helpful to examine how conserved these residues are in a much larger alignment of C-domains, grouped according to their acceptor residue specificities. At present, the authors cite reference 36 to support the generality. However, ref 36 specifically states that acceptor site specificity is a problem to domain substitution and therefore implies this will not be a general approach beyond the closely related substrates or C domains tested by the current authors. It would be helpful if the authors could test the other six Cs domains displayed in Figure S4a.

Lines 293 to 295: It is unclear that this work "fills a gap" per se - rather, the authors are contributing additional knowledge. As already noted, the authors previously acknowledge that ref 12 already shows Cs domains can be engineered to alter the identity of the preferred lipo substrate.

Lines 299 and 300: As noted above (lines 225-235), substantially more data is needed to support this conclusion. The recommended data from the other Cs domains from Figure S4a would help.

Lines 305 to 307: Again, the XU work referred to here specifically states that it is a rule to respect acceptor site specificity of the C domain. The authors' statements that their proposal is supported by this work are not correct.

Lines 308 to 318: This feels like an over-claim. The substrate binding pocket is presented in ref 12 based on modelling and in vitro experiments, and the current manuscript appears to have used this prior work as a basis for their point mutations of the Cs domain. More careful phrasing is required to highlight the novelty of the authors' work without stretching too far. This paragraph should also address the discrepancies between the in vitro and in vivo findings - how predictable are the outcomes of Cs domain bioengineering likely to be?

Supplementary Figures 1b and 1d. The peaks in the chromatograms could be anything and it is not at all clear that these are compounds consistent with 5a and 7. Comparison to standards, or the EIC or spectra is needed here.

Supplementary Figures S2b and S3b. Some peaks are annotated but are indistinguishable. For instance, the first peak related to Thr is not visible for compound 1, or the peaks at approx. 142 and 603 for compound 2a. Were these peaks actually detected, or were the annotations added automatically?

Supplementary Figure S2b. The peaks for the last Ala appears to be incorrect. The mass difference between the two peaks is approx 54, whereas it should be 71 based on the location of the dashed lines shown on the chemical structure.

Supplementary Figure S8: This figure needs comparison to a positive control, as low levels of activity may be detected from essentially non-active enzymes. For instance, Ehmann et al ([https://doi.org/10.1016/S1074-5521\(00\)00022-3](https://doi.org/10.1016/S1074-5521(00)00022-3)) similarly used amino-acyl SNACs to show strict C domain specificity towards acceptor substrates, yet were still able to detect condensation at low levels by non-cognate substrates.

Figures in general. The number of replicates MUST be stated throughout. Many appear to be a single replicate which raises repeatability concerns.

The document needs comprehensive proof-reading as the language is unclear at times. Below is a very small collection of representative examples.

- Line 32: Typo NPRSs
- The sentences from lines 46 to 57 are unclear
- Line 48: Should it say "C domains"?
- Line 153: The multiple uses of "confirmation" should be "conformation"
- Figure 1 legend. Should the following sentence say "or" and "moiety"? - "bearing a short acetyl and a medium octanoyl moieties"
- Figure 5f: Yield is incorrectly spelt on the y-axis.
- Line 184: "Alternation" is not correct
- Line 382: Should be "incubated", i.e. continually incubated for another day.
- Line 386: Should be "linear" not "liner"
- Line 418: Should be "described" not "descripted"

- David F Ackerley

Reviewer #2 (Remarks to the Author):

The manuscript by Zhong and colleagues presents the structural and functional analysis of an initiating condensation domain from the lipopeptide rhizomide. Herein the authors examine the starter condensation (Cs) domain of rhizomide A and holrhizin A NRPS clusters, which incorporate differing acyl chains, and illustrate the ability to swap the Cs domains, structurally characterize the Cs domain of the RzmA protein, and use the structure to guide further mutagenesis experiments.

The authors demonstrate with in vitro biochemical reactions that the Cs domains of each protein catalyze the transfer of the cognate acyl moiety from acyl-CoA to the SNAC-amino acid of the first encoded amino acid. The genetic construction of chimeric NRPS genes by swapping the full Cs domains, but not the N-terminal subdomain, then enabled the production in *S. brevistalis* of rhizomide A and holmyzin A analogs with the swapped preference of the acyl chain. In comparison with the existing structure of the CDA condensation domain, the authors chose a residue for mutation that may influence the acyl specificity. A series of mutations were made that correlated nearly perfectly with the size of the residue and the length of the acyl chain that was incorporated, although the rhizomide system showed a persistent ability to favor production of the C2 (acetyl) variant, perhaps related to the higher concentration of AcCoA in the cell. The authors may wish to comment on whether other changes were attempted or if this single residue was identified through the sequence/structure comparisons and resulted in such a clean specificity profile. The authors then determined the structure of the RzmA Cs domain in the unliganded state. The structure of a mutant (R148A) bound to octanoyl-CoA was also determined, illustrating a more closed configuration of the two subdomains. This structure guided additional mutational analysis of the acyl chain specificity.

The authors then captured a ternary complex with an acyl-CoA donor, the Leu-SNAC acceptor, utilizing a catalytically deficient mutant by mutating a conserved His residue. This gives them a structure of a completely closed, presumably catalytic conformation, if the His residue were intact. And finally, the authors determine another unliganded structure which gives a more closed conformation, although two loops at the active site are configured in an open state that would

allow a product to exit the active site pocket. The authors refer to this as the “product-released” state. This leads the authors to conclude a three-state conformational cycle wherein the two subdomains of the Cs domain close over the two substrates, open active site loops to release product, and finally open the two subdomains to re-adopt the fully open state.

This is a very nice study that provides insights into the binding of both the donor and acceptor ligands in the active site of an initiating condensation domain. The structure/function analysis is very nicely tested in mutants that expand the active site to allow different acyl chains to bind and react.

This three-state conformational cycle is overstated. Particularly because the crystals from this latest form derive from protein that was grown in the presence of both ligands and because the crystals took two weeks to grow—appearing as a second crystal form—it is possible that the initial protein crystals grew in the closed conformation. The catalytic mutant may still retain some degree of activity to catalyze the reaction, resulting in product release. I think rather what the structures demonstrate is that the active site loops can open independently of the opening of the two lobes of the Cs domain. This means that product release may arise from simply opening the floor and lid loops. But the results cannot rule out that in the context of the natural catalytic cycle, the opening of the two lobes precedes or accompanies the opening of the active site loops that, in the current experiment, only occur because the protein is constrained by the crystal lattice. The authors should reconsider this possibility and revise their presentation and discussion of the three-state model.

The other significant concern is the authors conclusion that the Cs domains use an Acyl-CoA rather than an Acyl-ACP as the lipid donor. This is based on the lack of a dedicated Acyl-CoA ligase or ACP in the cluster and the fact that catalytic activity is biochemically observed with acyl CoA. However, the crystal structure shows lower quality density for the nucleotide portion of the hexanoyl-CoA (Figures S6 and S7) compared to the pantetheine. Further, it does not appear that the authors ever tested an Acyl-ACP as a substrate. Thus, they may wish to soften the conclusion (lines 81-82) that the Acyl-CoA is the correct *in vivo* substrate. Acyl-ACPs might be pulled from primary metabolism for use.

Minor comments.

Throughout, the authors use apo to describe the unliganded structures. Technically, an apo enzyme refers to the absence of a necessary cofactor, although it is often used incorrectly to refer to a protein bound to no substrates, products, or other other small molecule analogs.

Line 77. “rarely short” should be revised to read “uncommon acetyl chain” or similar.

Lines 112-116. Please include here that this observation occurred with the *in vivo* production system.

Line 137. Revise to “asymmetric unit”

Line 146. co-crystallize

Line 185. Revise “allowed to greatly change”

Line 203. “conformations” This correction is also necessary in figures S6 and S10, which use confirmation in several places.

Line 224. Replace “approximated” with “near”, or similar.

Lines 284, 301, and 302. The use of lipo in place “acyl chain” is unconventional.

Line 285. Perhaps replace “altering supplement” with “supplementing with alternate donor substrates ...”

Line 291, "were below expectations" is a bit ambiguous. Perhaps clearly state that yield was low or other suboptimal results.

Line 316-317. The opportunity to alter the Cs specificity might be better described as a complement to attempts to alter the peptide backbone.

Line 324. Perhaps replace "confirmed" with "supported"

Line 326. Replace "conversed" with "conserved"

Line 341. Replace "shorts" with "shots"

Line 435. Replace "sequences" with "sequence"

Line 463. Are substrate concentrations correct as one is mM and other μ M?

Andrew M. Gulick
University at Buffalo

Reviewer #3 (Remarks to the Author):

The manuscript provides a large amount of data examining the structure, function and engineering of starter NRPS C domain that install acyl groups. Significantly, the authors established an approach to use domain swapping in combination with amino acid mutagenesis to rationally alter the substrate specificity both in vivo and in vitro. The protein crystallography is also a strength, multiple structures and complex structures provide novel insight into the function, although I think some of the interpretations are overstated. There are many grammatical errors and incorporation of incorrect English phrases in the text, making it hard to interpret the results/manuscript at times.

Points to consider:

- I don't think by common criteria (chemical, biochemical) one would classify acetic acid (acetate, acetyl) as a fatty acid or an acetyl-peptide as a lipopeptide. Also, 'lipo' as a standalone noun is not a standard abbreviation in the field.
- For the in vivo experiments (line299-300), it is not clear if the mutations were made on the hybrid C domain or WT.
- There is an issue of generality/utility of the results, as the domain swaps were done on homologous domains/systems from the same organism.
- I think the claims of assigning function to the core C domain motif is overstated. The roles of the amino acids have been assigned in other NRPS C domains, (although not starter) is many publications.
- I also think the claims of visualizing dynamics of the catalytic cycle are overstated, especially the 'product release' structure. The subdomains are known to be flexible and this conformation is a result of crystal contacts, there is no supporting evidence that this is a conformation along the catalytic cycle

Author's Response to Reviewer #1:

Overview

The structures and rational mutagenesis studies presented in the manuscript submitted by Lin Zhong and colleagues are really very interesting, and likely to hold substantial interest for others in the field. However, there are some inconsistencies between some of the experiments that need to be discussed in greater detail. Some of the conclusions are also a bit too broad, given the focus on just two enzymes, and would require further experimental data to substantiate them. Additional explanations are needed to clarify some of the work, and there is a strong need for a professional proof-reading service – nearly every sentence contains typographical errors and/or grammatical incongruities, and in some places these greatly obscure the intended meaning.

Response: Thank you very much for your kind comments and constructive suggestions. We performed new experiments to address the inconsistencies, with the data and related descriptions added in the manuscript (see below detailed responses). We conducted full-length Cs domain swapping to successfully change the fatty acyl chain of glidobactin A, showing the generality of this approach. We carefully changed some contexts and conclusion to appropriately state our results and the significance according to the suggestions, and the revised manuscript was then extensively edited by an English native-speaker scientist and the Nature Research Editing Service (certificate attached), to improve the grammar, scientific accuracy, clarity and instructiveness of our manuscript.

Specific comments

Lines 87 to 89. Raw numbers of percentage identity are not overly informative. An alignment of these two Cs domains would be helpful to have in the supplementary materials and could also be used to exemplify the recombination points used for Cs domain and subdomain substitution (see next point).

Response: We added the protein sequence alignment of RzmA-Cs, HolA-Cs and GlbF-Cs domains in Fig. 1d, which are used for domain swapping in this study. The accurate sites of (sub)domain swapping were labeled in Fig. 1d, the full-length swapping of Cs domain includes the C-A linker region. The swapping sites of the full-length Cs domain and subdomain (N-lob) are labeled with a black line and a pair of scissors in Fig. 1d.

Lines 89 to 92. Very little information was provided as to the rationale for selecting subdomain recombination points. I request that the location be noted precisely in the alignment suggested above – this would greatly aid the reader.

Response: As mentioned above, we added the sequence alignment and annotated the swapping sites in Fig. 1d.

Lines 97 to 99. Given the sample size of two, and that only one recombination point was tested, there is very little evidence to conclude that Cs domain substitution is more reliable than subdomain substitution. More substitutions employing Cs domains from a range of different NRPSs and several different recombination points would be needed to make such a statement with any confidence.

Response: Thanks a lot for your suggestions. We softened the statement (lines 105-108 in the revised version) and meanwhile complemented Cs (sub)domain swapping experiments in another biosynthetic gene cluster to extend its application (lines 109-128 in the revised version, Fig. 3a, b).

The original statement was changed to "The subdomain (N-terminal or N-lobe) swapping of the Cs domain also yielded the expected products (**1d** and **2a**) but in a relatively lower yield and conversion ratio (Fig. 2c, d), suggesting that Cs domain swapping would be a feasible approach to change the acyl chains of lipopeptides." in lines 105-108.

We swapped the Cs domain of glidobactin biosynthetic gene cluster in the original producer *S. brevitalea* DSM 7029 with RzmA-Cs, HolA-Cs and GlpC-Cs to construct the hybrid gene clusters. The exchange of HolA-Cs and GlpC-Cs domains changed the original unsaturated 2(*E*),4(*E*) dodecadienoyl of glidobactin A to saturated octanoyl (C8) and decanoyl (C10) chains with improved yields, respectively (Fig. 3a,b, NMR: Table S7, Figs. S29-S32). However, the change to RzmA-Cs failed to produce any products, possibly due to the acceptor specificity of the RzmA-Cs domain and/or the donor specificity of the downstream Glb C domains. Subdomain (N-lobe) swapping of GlbF-Cs with that of GlpC-Cs was also conducted, but no expected compounds were detected. Combined with the swapping in RzmA and HolA (Fig. 2), we propose that full-length Cs domain swapping could be a worthy choice to modify the acyl chains of lipopeptides, at least for these three lipopeptides. The results were presented in the results section (lines 109-128 in the revised version and Fig. 3a-3b), and the NMR data of C8-glb (**7b**) and C10-glb (**7c**) for structural elucidation were shown in Table S7, Figs. S29-S32. These data suggested the potential generality of this approach to change the acyl chains.

Lines 100 to 102. It is already known that the Cs domain can define specificity of lipid chains - indeed, the authors specifically mention that this is known in lines 42 to 46. However, this sentence now makes it sound as if the authors are claiming that they have observed this as a new discovery. In addition, ref 12 cited in this manuscript already shows that it is feasible to alter Cs domain donor site specificity with point mutations. Thus, the authors need to tone down their claims and put them in context so that it does not sound like they are seeking credit for discovering previously-known phenomena.

Response: Thanks for your comments. To soften the claim, we decided to delete the sentence "Thus Cs domain crucially defines the selectivity of lipid chains, and it is a feasible target for the engineering of lipopeptide-producing". In the study of Ref 12, as pointed out, they indeed showed that the feasibility to alter Cs domain donor site specificity with point mutations by modeling Lpta-C1 structure and the *in vitro* experiment using relatively narrow donor substrates (C8-C12). In our results shown in this paragraph (lines 94-128), we used the Cs domain swapping to change the acyl chains of lipopeptides *in vivo*, providing direct examples for the claim that Cs domain is a feasible target for the engineering of Cs-contained NRPS to producing novel lipopeptides, which is a more in-depth study. We also compared the ref 12 and our results in the discussion section (lines 341-347): "A recent study using modeling and *in vitro* experiments identified four residues as functionally related to the fatty acyl substrate selectivity of LptA-C112, but the residues were different from our revealed key sites demonstrated by the cocrystal structure and by *in vitro* and *in vivo* experiments. We found that three sites in RzmA-Cs Q36, Y138 and R148 play key roles in controlling the specificity of the acyl chain, in contrast to the previously study on LptA-C1 in A54145 biosynthetic pathway¹², i.e., A152, A369, A386 and L397 (corresponding to Y149, D350, N367 and Y378 in RzmA-Cs) (Fig. S4a)."

*Lines 116 to 119. Figures 3c and S1e do not really seem to be in agreement, as the authors claim here. Figure 3c shows the major products for the R148G variant to contain a lipid with n=0 and n=6, whereas the major products for the R148G variant in Figure S1e are predominantly lipids with n=6 and n=8. Was R148G tested with all of 5a to 5e, or only 5d/5e? Or is there some greater inconsistency between the *in vivo* experiments and *in vitro* assay? A more in-depth discussion is required here, and the legend to Figure S1 requires substantially more detail, to explain exactly what was tested, and how.*

Response: In Fig. S1e, the R148G was tested with C2 (n=0), C4 (n=2), C6 (n=4), C8 (n=6) and C10 (n=8) -CoAs (**3a** to **3e**) to get the possible products **5a** (n=2) to **5e** (n=8). We remade the Fig. S1e and added columns with undetectable product to clearly show the products of each mutation. The legend

of Fig. S1e was also modified.

The RzmA-Cs R148G mutant yielded products including C2-RzmA (n=0) and C8-RzmA (n=6) *in vivo*, but could not produce C2-Leu-SNAC (n=0) *in vitro*. We thought that this inconsistency was due to complex levels of this two systems between *in vivo* and *in vitro*, such as differences in substrates concentrations, reaction times etc. We tried to simulate the *in vivo* conditions as much as possible by extending reaction time and adjusting concentrations of donor substrates in the *in vitro* assay. The products C2-Leu-SNAC and C4-Leu-SNAC were detected with relatively high yields *in vitro*, as shown by the counterparts in the *in vivo* experiments (Fig. S10). As we discussed this inconsistency in the discussion section (lines 347-359): "We also noticed the inconsistencies between our *in vivo* and *in vitro* experiments due to the complexity of the *in vivo* system or to different products. The *in vitro* experiments determined only the formation of the first simple biosynthetic intermediate mimics by a Cs domain-catalyzed reaction, while the *in vivo* experiments showed the final complex products after many reactions. The concentrations of the acyl-CoA substrates vary widely in cells; e.g., acetyl-CoA, as an important primary metabolite, is present at concentrations at least one or two orders of magnitude higher than those of other medium or long fatty acyl-CoAs in *E. coli* (Ref 44). We set up another *in vitro* experiment to simulate *in vivo* conditions using higher concentrations of acetyl- and butanoyl-CoAs (C2-CoA: C4-CoA: C6 to C18-CoA=1000:100:1) and an extended reaction time, and the products C2-Leu-SNAC and C4-Leu-SNAC were detected with relatively high yields, as shown by the counterparts in the *in vivo* experiments (Fig. S10).

We added description in Fig. S1e legend" The donor substrates contains same concentrations of C2-, C4-, C6-, C8 and C10-CoAs (3a-3e) were reacted with mimic substrates of acceptor L-Leu-SNAC for each Cs domain variants." More detailed information was shown in Enzymatic activity assay of Method section.

Figures 5e and 5f. Again, there appear to be inconsistencies between the in vivo and in vitro data that the authors do not acknowledge. For example, the major product in vivo is frequently the C2-Rzm, but this is not observed at all for most of the variants in vitro. Were all substrates tested in all cases? The graphs appear to be drawn in R and the positioning of the bars suggests that either not all substrates were tested, or some data has been omitted. As per my previous comment, the legend requires substantially more detail to clearly explain how each set of experiments was performed. Please also note that the choice of colour scheme makes it very difficult to judge differences between C2-Leu-SNAC and C8-Leu-SNAC in 5f.

Response: For Fig. 5f (now Fig. 7b), the *in vitro* substrate competition experiment was performed using same concentrations of each donor substrate mixture (from C2-CoA to C18-CoA) in a relatively short time. As we mentioned

above, we simulated the *in vivo* condition as much as possible and discussed this inconsistencies in lines 346-358 and Fig.S10. We also added in method section (lines 550-552) "For the substrate competition assay, we used the mixture of 80 μ M of each acyl-CoA as donor substrates to investigate the specificities of RzmA-Cs and its variants for acyl-CoAs. The relative yield of each product was determined by comparison of its peak area in the UPLC-MS chromatogram." We also redraw the graph to arrange bar for every product including that could not be detected (ND, *) and change the column colors for clear presentation and easy judgement (now Fig. 7a, b), as we mentioned in Fig.7 legend" The products that were not detected (ND) in UPLC-MS were marked with asterisks."

Figures 5e and 5f. Another inconsistency that is sufficiently prominent I have noted it as a separate point - why is it that yield of the C2 product was so low for the R148 WT in Figure 5f, when it was one of the highest yields in vivo? Conversely, why are mutants that had very low yield in Figure 5e suddenly high yielding in Figure 5f?

Response: As we mentioned above, the *in vivo* and *in vitro* systems were different. For the mutants with higher production *in vitro* compared to *in vivo*, a similar question has been responded above (lines 347-359). In the *in vitro* assay, the concentrations of each acyl-CoA were equal, but their concentrations vary greatly *in vivo* (Ref. 44). The low yield of the C2 product of the R148 WT in Fig. 5f (now Fig. 7b) was caused by the relatively low concentration of acetyl-CoA in the *in vitro* assays compared to the *in vivo* condition, in which the concentration of acetyl-CoA is at least one or two orders of magnitude higher than those of other medium or long fatty acyl-CoAs. The concentrations of medium or long acyl-CoAs are relatively high *in vitro*, led to the suddenly high yields of their products in Fig. 5f (now Fig. 7b). Although some differences existed within these two systems, the data showed similar trends for most mutations. The inconsistency did not affect our conclusion that these residues play the key role in controlling donor specificities. This explanation is also in agreement with the comment of reviewer #2 "perhaps related to the higher concentration of AcCoA in the cell."

Figure 3c and 5e: R148G is analysed in vivo in both Figures 3c and 5e. In 3c the major products are $n = 0$ and $n = 6$, with no trace of $n = 8$; however, in 5e the major products have changed to be $n = 0$ and $n = 8$, with no trace of $n = 6$. It is unclear what difference there is between these experiments.

Response: Thanks for your kind comments. For R148G in Fig. 3c, C2-RzmA ($n=0$) and C8-RzmA ($n=6$) were produced. In Fig. 5e (now Figure 7a), same

products C2-RzmA (n=0) and C8-RzmA (n=6) were generated in R148G. This was a mistake for “n=8 means C8-RzmA” (Figure 1b), and it should be “n=6 for C8-RzmA”. We verified all products (names and numbers) and added more information of compounds in both figures to make it easier to understand and follow.

Lines 160 to 163: Does mere proximity suggest the histidine is involved in substrate positioning? Is the substrate orientation consistent with this hypothesis and is there any hydrogen bonding supporting the positioning role? These assumptions do receive some later experimental support when this histidine is substituted to a valine (lines 196-200), but at this earlier stage of the text, providing additional guidance to the reader would be useful.

Response: Yes, the close distance (3.6 Å) suggested that the histidine is involved in substrate positioning. The acceptor substrate orientation is consistent with the previous report (Ref. 19), we here also show the donor substrate positioning from the cocrystal structure (lines 186-193). And this inference was verified by experimental data further, as we mentioned in lines 219-246. We also rephrased this sentence to make it clear (lines 186-189): “Interestingly, the ε nitrogen of H140 (the second histidine of the “HHxxxDG” motif) is relatively close to the acyl group of the donor substrate (3.6 Å), suggesting that the substrate positioning function of this histidine applies not only to the acceptor substrate as inferred previously (Ref. 19) but also to the donor substrate (the C8-CoA in this Cs).”

Lines 225 to 235: The authors identify putative substrate-determining residues and make the argument that these might be relevant for C domains with different acceptor substrates. However, Figure S4a does not explain which substrate each C domain accepts, which would be useful information. Indeed, to support their argument, it would be helpful to examine how conserved these residues are in a much larger alignment of C-domains, grouped according to their acceptor residue specificities. At present, the authors cite reference 36 to support the generality. However, ref 36 specifically states that acceptor site specificity is a problem to domain substitution and therefore implies this will not be a general approach beyond the closely related substrates or C domains tested by the current authors. It would be helpful if the authors could test the other six Cs domains displayed in Figure S4a.

Response: Thanks for your suggestions. We tested the *in vitro* donor substrate specificities of RzmA-Cs, HoIA-Cs and GIBF-Cs with seven SNACs, e.g., L-Val-SNAC, L-Leu-SNAC, L-Ala-SNAC, L-Phe-SNAC, L-Met-SNAC, L-Ser-SNAC and L-Thr-SNAC. All of them indeed showed relatively broad acceptor specificities (Fig.

3c, d in the revised version). The Figure S4a just showed the key amino acid residues in these Cs domains. A large alignment of Cs domain was performed, but we cannot group them according to their acceptor residue specificities. We modified the description in this paragraph mentioned in lines 249-262 "In addition, sequence alignment of several other Cs domains with same or different substrates revealed no strict rule of conservation for the above three residues, except that vast majority of them are short-chain residues (Fig. S4a). The above *in vitro* experiments of RzmA-Cs, HolA-Cs and GlbF-Cs domains also showed relatively wide specificities for donor substrates (Fig. 3c, d). All these above results correlate well with the fact that Cs domains could accommodate certain degree of variation for the "acceptor" substrates, while the neighboring A domains are more responsible for the substrate selectivity. This finding also supports the Cs domain-swapping strategy to change the acyl chains (Fig. 2c,d), and evidences the NRPS module-swapping at the C-A linker region for combinatorial biosynthesis of nonribosomal peptides (Ref. 36, 38)."

We cited Ref. 36 (Ref. 38 in the revised version, Bozhuyuk, et al. Nat Chem, 2018) here just to prove the C-A linker region could be a good site for modular/domain swapping for combinatorial biosynthesis of NRPs. Our results support this strategy (Ref 38 in the revised version) and also the very recent study to change the A domain (Ref. 36 in the revised version, Calcott, et al, Nat Commun, 2020). Of course, the acceptor site specificity should not be ignored, because as we know NRPS is an assembly line system, the specificities of all domains could exist. We also softened our statement in the manuscript. We will try to group the C domains according to their acceptor residue specificities if possible in the future, which may need more bioinformatic analysis that may beyond our capability.

We performed the donor substrate specificities of three of six Cs domains in Figure S4a. The results showed relatively broad acceptor specificities (Fig. 3c, d in the revised version) as shown above.

Lines 293 to 295: It is unclear that this work "fills a gap" per se – rather, the authors are contributing additional knowledge. As already noted, the authors previously acknowledge that ref 12 already shows Cs domains can be engineered to alter the identity of the preferred lipo substrate.

Response: Thanks for your kind suggestion. To remove the misunderstanding, we changed this sentence to "Here, we changed the acyl chains of nonribosomal lipopeptides by bioengineering of Cs domain in two ways, Cs domain swapping and point mutation." in lines 322-323. As for Ref.12, we discussed this issue as noted above.

Lines 299 and 300: As noted above (lines 225-235), substantially more data is

needed to support this conclusion. The recommended data from the other Cs domains from Figure S4a would help.

Response: We think this claim is overstated, although we complemented the donor substrate specificities of three of six Cs domains in Figure S4a (as mentioned above). So we deleted this sentence, and rewrote this paragraph lines 327-332.

Lines 305 to 307: Again, the XU work referred to here specifically states that it is a rule to respect acceptor site specificity of the C domain. The authors' statements that their proposal is supported by this work are not correct.

Response: Thank you very much for your correction. We deleted the sentence and rewrote several sentences in lines 334-337 "The recently reported research on the modification of NRPSs by adenylation domain (A domain) substitution alone also supported that C domain specificities were not as strict as previously thought(ref36). Our results suggested that the use of the full-length Cs domain as a swapping unit is a feasible way to modify lipopeptides."

Lines 308 to 318: This feels like an over-claim. The substrate binding pocket is presented in ref 12 based on modelling and in vitro experiments, and the current manuscript appears to have used this prior work as a basis for their point mutations of the Cs domain. More careful phrasing is required to highlight the novelty of the authors' work without stretching too far. This paragraph should also address the discrepancies between the in vitro and in vivo findings – how predictable are the outcomes of Cs domain bioengineering likely to be?

Response: In our study, we used our cocrystal structure in this manuscript (Figure 6) as a basis for our point mutation of Cs domain, which is independent on the modelling and *in vitro* experiments in Ref 12. The novelty and impact of our cocrystal structures and mutations were judged very favorably by reviewer #2.

We discussed the Ref. 12 in lines 341-347 in the revised version of this manuscript. The discrepancies between the *in vitro* and *in vivo* findings were also discussed in lines 347-359 (Fig. S10) as mentioned above.

Supplementary Figures 1b and 1d. The peaks in the chromatograms could be anything and it is not at all clear that these are compounds consistent with 5a and 7. Comparison to standards, or the EIC or spectra is needed here.

Response: Thanks for your suggestion. We added the EICs in the legend of

Figs. S1b, S1d accordingly.

Supplementary Figures S2b and S3b. Some peaks are annotated but are indistinguishable. For instance, the first peak related to Thr is not visible for compound 1, or the peaks at approx. 142 and 603 for compound 2a. Were these peaks actually detected, or were the annotations added automatically?

Response: All m/z values were shown automatically from Bruker Compass Data Analysis software, but some values were not shown automatically in this spectra due to the low intensities. We added some lines and adjusted the positions of values to make it clearer. All assigned m/z values of ion peaks were automatically shown after zoomed in the MS spectra. We remade the Figures S2 and S3 to clarify the data. The 142 for compound **2a** was not showed clearly, so we did not give numbers in MS spectra. Compound **2** is a known compound that we reported previously (Wang, et al. PNAS, 2018. Ref 29), and **2a** was also purified for NMR recording, as mentioned in Fig. S11-S15 and Table S6.

Supplementary Figure S2b. The peaks for the last Ala appears to be incorrect. The mass difference between the two peaks is approx 54, whereas it should be 71 based on the location of the dashed lines shown on the chemical structure.

Response: Thanks for your careful check. It's our labeling mistake, and we corrected it to 71 based on the automatic annotations created by Bruker Compass Data Analysis software.

Supplementary Figure S8: This figure needs comparison to a positive control, as low levels of activity may be detected from essentially non-active enzymes. For instance, Ehmann et al ([https://doi.org/10.1016/S1074-5521\(00\)00022-3](https://doi.org/10.1016/S1074-5521(00)00022-3)) similarly used amino-acyl SNACs to show strict C domain specificity towards acceptor substrates, yet were still able to detect condensation at low levels by non-cognate substrates.

Response: This figure was used to show the acceptor specificity of RzmA-Cs domain to non-cognate substrate L-Val. Because we have supplied more experiment data and make a new figure (Fig. 3c,d) in the main text (lines 117-120 and lines 250-259) to show the acceptor specificities of Cs domain using multiple aminoacyl-SNACs including the native and non-cognate substrates, as mentioned above, the original Fig S8 was deleted, and changed to a new figure (Fig. 3c, d).

Figures in general. The number of replicates MUST be stated throughout. Many

appear to be a single replicate which raises repeatability concerns.

Response: Thanks for your kind suggestion, we mentioned this sentence "The experiments were conducted in triplicates. Error bars, SD; n = 3." in related figure legends and/or also the method section.

The document needs comprehensive proof-reading as the language is unclear at times. Below is a very small collection of representative examples.

- *Line 32: Typo NPRSs*

Response: changed to "NRPSs"

- *The sentences from lines 46 to 57 are unclear*

Response: We rephrased these sentences (lines 52-62) in the revised version of this manuscript) to "The crystal structure of the dissected Cs domain from CdaPS1 enabled researchers to propose an active-site tunnel for the accommodation of both "donor" and "acceptor" substrates (Ref. 18). Due to the low affinity of C domains for substrate analogs, a covalent mimic of an acceptor substrate was adopted to obtain the cocrystal structure of CdaPS1-Cs, which revealed the role of second histidine (of the "HHxxxDG" motif) in substrate positioning and the acceptor substrate specificity of the condensation reaction(Ref.19). Although a number of NRPS structures involving the C domains have been solved as single-domain (Ref. 18-20) or multidomain (even modular) structures (Ref. 21-28), CdaPS1-Cs remains the only lipoinitiation-conducting Cs domain with a reported structure. Moreover, very few C domain structures were solved in complex with substrates; in particular, none were solved with the acyl moiety. These setbacks hinder the revealing of the lipoinitiation mechanism and engineering of the acyl chain of lipopeptides."

- *Line 48: Should it say "C domains"?*

Response: Changed to "C domains"

- *Line 153: The multiple uses of "confirmation" should be "conformation"*

Response: Changed to "conformation" through the manuscript.

- *Figure 1 legend. Should the following sentence say "or" and "moiety"? - "bearing a short acetyl and a medium octanoyl moieties"*

Response: Changed to "bearing a short acetyl or a medium octanoyl moieties"

- *Figure 5f: Yield is incorrectly spelt on the y-axis.*

Response: Changed accordingly

- *Line 184: "Alternation" is not correct*

Response: Changed to "alteration" in line 210

- *Line 382: Should be "incubated", i.e. continually incubated for another day.*

Response: Changed to "incubated"

- *Line 386: Should be "linear" not "liner"*

Response: Changed to "linear"

- *Line 418: Should be "described" not "descripted"*

Response: Changed to "described"

- David F Ackerley

Author's Response to Reviewer #2:

The manuscript by Zhong and colleagues presents the structural and functional analysis of an initiating condensation domain from the lipopeptide rhizomide. Herein the authors examine the starter condensation (Cs) domain of rhizomide A and holrhizin A NRPS clusters, which incorporate differing acyl chains, and illustrate the ability to swap the Cs domains, structurally characterize the Cs domain of the RzmA protein, and use the structure to guide further mutagenesis experiments.

*The authors demonstrate with in vitro biochemical reactions that the Cs domains of each protein catalyze the transfer of the cognate acyl moiety from acyl-CoA to the SNAC-amino acid of the first encoded amino acid. The genetic construction of chimeric NRPS genes by swapping the full Cs domains, but not the N-terminal subdomain, then enabled the production in *S. brevitelia* of rhizomide A and holmyzin A analogs with the swapped preference of the acyl chain. In comparison with the existing structure of the CDA condensation domain, the authors chose a residue for mutation that may influence the acyl specificity. A series of mutations were made that correlated nearly perfectly with the size of the residue and the length of the acyl chain that was incorporated,*

although the rhizomide system showed a persistent ability to favor production of the C2 (acetyl) variant, perhaps related to the higher concentration of AcCoA in the cell. The authors may wish to comment on whether other changes were attempted or if this single residue was identified through the sequence/structure comparisons and resulted in such a clean specificity profile. The authors then determined the structure of the RzmA Cs domain in the unliganded state. The structure of a mutant (R148A) bound to octanoyl-CoA was also determined, illustrating a more closed configuration of the two subdomains. This structure guided additional mutational analysis of the acyl chain specificity.

The authors then captured a ternary complex with an acyl-CoA donor, the Leu-SNAC acceptor, utilizing a catalytically deficient mutant by mutating a conserved His residue. This gives them a structure of a completely closed, presumably catalytic conformation, if the His residue were intact. And finally, the authors determine another unliganded structure which gives a more closed conformation, although two loops at the active site are configured in an open state that would allow a product to exit the active site pocket. The authors refer to this as the "product-released" state. This leads the authors to conclude a three-state conformational cycle wherein the two subdomains of the Cs domain close over the two substrates, open active site loops to release product, and finally open the two subdomains to re-adopt the fully open state.

This is a very nice study that provides insights into the binding of both the donor and acceptor ligands in the active site of an initiating condensation domain. The structure/function analysis is very nicely tested in mutants that expand the active site to allow different acyl chains to bind and react.

This three-state conformational cycle is overstated. Particularly because the crystals from this latest form derive from protein that was grown in the presence of both ligands and because the crystals took two weeks to grow—appearing as a second crystal form—it is possible that the initial protein crystals grew in the closed conformation. The catalytic mutant may still retain some degree of activity to catalyze the reaction, resulting in product release. I think rather what the structures demonstrate is that the active site loops can open independently of the opening of the two lobes of the Cs domain. This means that product release may arise from simply opening the floor and lid loops. But the results cannot rule out that in the context of the natural catalytic cycle, the opening of the two lobes precedes or accompanies the opening of the active site loops that, in the current experiment, only occur because the protein is constrained by the crystal lattice. The authors should reconsider this possibility and revise their presentation and discussion of the three-state model.

Response: Many thanks for all the above positive comments and instructive suggestions. We agree that the “three-step model” we previously proposed is not very accurate. The three structures with different conformations are representative snap-shots captured during the reaction cycle. We have revised the descriptions accordingly in the manuscript.

We revised the presentation and discussion of the three-step model as “Based on these H140V/R148A structures and related functional assays, we propose three potential conformational states (i.e., “unbound-bound-released”) within the lipoinitiation reaction cycle catalyzed by Cs domains (Fig. 9d, also illustrated in Supplementary Video 1).” in lines 286-289 of results section. And “Fortunately, we were able to capture several snap-shots of the conformations of RzmA-Cs, illustrating a potential reaction cycle including the “unbound-bound-released” states (Supplementary Video 1).” in lines 390-392 of discussion sections.

The other significant concerns is the authors conclusion that the Cs domains use an Acyl-CoA rather than an Acyl-ACP as the lipid donor. This is based on the lack of a dedicated Acyl-CoA ligase or ACP in the cluster and the fact that catalytic activity is biochemically observed with acyl CoA. However, the crystal structure shows lower quality density for the nucleotide portion of the hexanoyl-CoA (Figures S6 and S7) compared to the pantetheine. Further, it does not appear that the authors ever tested an Acyl-ACP as a substrate. Thus, they may wish to soften the conclusion (lines 81-82) that the Acyl-CoA is the correct in vivo substrate. Acyl-ACPs might be pulled from primary metabolism for use.

Response: Thank you very much for the kind suggestion. We softened the conclusion and rephrased to “Thus, we infer that the Cs domains of RzmA and HolA may catalyze lipoinitiation using acyl-CoAs as direct substrates, but the possibility of harnessing acyl-ACPs as direct substrates cannot be fully excluded, because the Cs domains might capture acyl-ACPs from the primary metabolism of bacteria.” in lines 87-90.

Minor comments.

Throughout, the authors use apo to describe the unliganded structures. Technically, an apo enzyme refers to the absence of a necessary cofactor, although it is often used incorrect to refer to a protein bound to no substrates, products, or other other small molecule analogs.

Response: We changed “apo” to “unbound” or “unliganded”, according to the contexts.

Line 77. "rarely short" should be revised to read "uncommon acetyl chain" or similar.

Response: Changed to "uncommon acetyl chain"

Lines 112-116. Please include here that this observation occurred with the in vivo production system.

Response: We rephrased this sentence "We found that compared with wild-type (WT) RzmA, mutations at R148 indeed altered the production profile of rhizomide *in vivo*" in line 140-141.

Line 137. Revise to "asymmetric unit"

Response: Revised to "asymmetric unit"

Line 146. co-crystallize

Response: Revised to "co-crystallize"

Line 185. Revise "allowed to greatly change"

Response: We rephrased this sentence "The triple mutant Y138A/M143A/R148G showed a large change in the specificity of the RzmA-Cs domain for acyl chains from short C2 to long C14, up to C16," in lines 211-213.

Line 203. "conformations" This correction is also necessary in figures S6 and S10, which use confirmation in several places.

Response: All revised to "conformations" accordingly.

Line 224. Replace "approximated" with "near", or similar.

Response: Replaced with "near"

Lines 284, 301, and 302. The use of lipo in place "acyl chain" is unconventional.

Response: Changed to "acyl chain"

Line 285. Perhaps replace "altering supplement" with "supplementing with alternate donor substrates ..."

Response: Revised to "relies on supplementing with alternate donor substrates based on..." in line 314.

Line 291, "were below expectations" is a bit ambiguous. Perhaps clearly state that yield was low or other suboptimal results.

Response: Revised to "However, the yields were low, as the tolerance of the Cs domain for donor and acceptor substrates is limited." in lines 320-321.

Line 316-317. The opportunity to alter the Cs specificity might be better described as a complement to attempts to alter the peptide backbone.

Response: Revised to "This Cs domain bioengineering strategy, including domain swapping and point mutation, is a complement to attempts to modify the peptidyl backbone in the field of combinatorial biosynthesis to produce novel lipopeptides and nonribosomal peptides." in lines 365- 367.

Line 324. Perhaps replace "confirmed" with "supported"

Response: Replaced with "supported".

Line 326. Replace "conversed" with "conserved"

Response: Replaced with "conserved"

Line 341. Replace "shorts" with "shots"

Response: Replaced with "shots"

Line 435. Replace "sequences" with "sequence"

Response: Changed to "sequence"

Line 463. Are substrate concentrations correct as one is mM and other μM ?

Response: It should be 8 mM aminoacyl-SNAC and 80 μM each acyl-CoA (C2 to C18-CoAs) in this assay, to make sure acyl-CoAs were maximumly reacted to form products by using excess aminoacyl-SNAC. We changed it in the Method section.

Andrew M. Gulick

University at Buffalo

Author's Response to Reviewer #3:

The manuscript provides a large amount of data examining the structure, function and engineering of starter NRPS C domain that install acyl groups. Significantly, the authors established an approach to use domain swapping in combination with amino acid mutagenesis to rationally alter the substrate specificity both in vivo and in vitro. The protein crystallography is also a strength, multiple structures and complex structures provide novel insight into the function, although I think some of the interpretations are overstated. There are many grammatical errors and incorporation of incorrect English phrases in the text, making it hard to interpret the results/manuscript at times.

Response: Thank you very much for your kind comments and constructive suggestions. We softened some interpretations and conclusion to appropriately state our results and the significance, according to all reviewer's suggestions. The revised manuscript was extensively edited by an English native-speaker scientist and the Nature Research Editing Service, to improve the English.

Points to consider:

- I don't think by common criteria (chemical, biochemical) one would classify acetic acid (acetate, acetyl) as a fatty acid or an acetyl-peptide as a lipopeptide. Also, 'lipo' as a standalone noun is not a standard abbreviation in the field.

Response: Thanks for your professional suggestion. We changed the "lipo, lipid chain and fatty acid" with "acyl chain or acyl" in the context to clarify the descriptions. We also added one sentence in lines 35-37 "Because the N-terminal acyls are highly diverse, ranging from short acetyl to long fatty acyl groups, we here mention the NRPS-derived peptide with an N-acylation as a nonribosomal lipopeptide for the purpose of a convenient description."

- For the in vivo experiments (line299-300), it is not clear if the mutations were made on the hybrid C domain or WT.

Response: All of point mutations were performed in the wild type of RzmA-Cs or HolA-Cs domains for *in vivo* experiments. To accurately interpret our results, the original sentence was deleted in the revised version of this manuscript as mentioned above, and the detailed discussion was shown in lines 338-367.

- There is an issue of generality/utility of the results, as the domain swaps were done on homologous domains/systems from the same organism.

Response: To test the generality of Cs domain swapping, we complemented Cs domain swapping experiments in the glidobactin gene cluster from *Schlegelella brevitalea* strain DSM 7029 using different Cs domains (lines 109-128 in the revised version, Fig. 3ab). The RzmA-Cs and HolA-Cs domains from *Paraburkholderia rhizoxinica* HKI 454, and GlpC-Cs domain from *S. brevitalea* DSM 7029 were used to replace the GlbF-Cs domain in glidobactin gene cluster. The exchange of HolA-Cs and Glp-Cs domains changed the original unsaturated 2(E),4(E) dodecadienoyl of glidobactin A to saturated octanoyl (C8) and decanoyl (C10) chains with improved yields, respectively (lines 109-128 in the revised version, Fig. 3ab). The GlbF-Cs domain was swapped by a heterologous HolA-Cs domain, which resulted in the successful production of C8-glidobactin, showing the Cs domain swapping can be achieved on heterologous domains from the different microorganism.

We also softened the conclusion "The subdomain (N-terminal or N-lobe) swapping of the Cs domain also yielded the expected products (**1d** and **2a**) but in a relatively lower yield and conversion ratio (Fig. 2c,d), suggesting that Cs domain swapping would be a feasible approach to change the acyl chains of lipopeptides." in lines 105-108. And added the sentence "we propose that full-length Cs domain swapping could be a worthy choice to modify the acyl chains of lipopeptides, at least for these three lipopeptides." in line 126-128.

- *I think the claims of assigning function to the core C domain motif is overstated. The roles of the amino acids have been assigned in other NRPS C domains, (although not starter) is many publications.*

Response: We agree that the function of the core C domain motif has been assigned by point mutations, biochemical experiments and crystallizations in the past two decades. In our study, our co-complex structures of RzmA-Cs suggested the positioning function of this histidine (H140) to the donor substrates, i.e., acyl-CoAs. In addition, we infer that the glycine within the motif (G145) is conserved, as a residue without the side-chain is able to make room for the donor substrates. Our study further confirmed the function of this motif in Cs domain for donor substrate positioning, and also provided the potential function of glycine in this motif. We softened the conclusion (lines 244-246) and also added more detailed discussion in lines 368-383.

- *I also think the claims of visualizing dynamics of the catalytic cycle are overstated, especially the 'product release' structure. The subdomains are known to be flexible and this conformation is a result of crystal contacts, there is no supporting evidence that this is a conformation along the catalytic cycle*

Response: Although we cannot fully rule out the possibility that the three

different conformations of RzmA-Cs we captured here depend on the crystal contacts, they may represent potential snap-shots during the reaction cycle. As for the “released” state, we provided evidences that the double mutant H140V/R148A was still able to catalyze the reaction but with a much lower efficiency (Fig. 9c). We softened the conclusion (lines 286-301) and revised the descriptions about the conformational changes in the manuscript accordingly (lines 385-398) as mentioned above.

REVIEWERS' COMMENTS

Reviewer #1 (Remarks to the Author):

I am satisfied that the authors have appropriately addressed all the points I raised in my previous review. I did note one small typographical error in their revised text in the legend to Figure 7 - the word "with" should be deleted from the phrase "for the with peak areas of EICs". Otherwise, I would just like to compliment them on their additional experimental work, which yielded some very nice results and have considerably strengthened several of their arguments.

David F Ackerley, Victoria University of Wellington, New Zealand

Reviewer #2 (Remarks to the Author):

This improved manuscript by Zhong et al has addressed my prior concerns. I am satisfied with the corrections and revisions.

Andrew Gulick

Reviewer #3 (Remarks to the Author):

The authors were able to address the concerns I expressed, specifically the issue of generality/utility of the presented results from additional experiments and conclusions.

Author's Response to Reviewer #1:

I am satisfied that the authors have appropriately addressed all the points I raised in my previous review. I did note one small typographical error in their revised text in the legend to Figure 7 – the word "with" should be deleted from the phrase "for the with peak areas of EICs". Otherwise, I would just like to compliment them on their additional experimental work, which yielded some very nice results and have considerably strengthened several of their arguments.

David F Ackerley, Victoria University of Wellington, New Zealand

Response: We thank this Reviewer for his remarks and compliment. We deleted the "with" in the legend of Figure 7 accordingly.

Author's Response to Reviewer #2:

This improved manuscript by Zhong et al has addressed my prior concerns. I am satisfied with the corrections and revisions.

Andrew Gulick

Response: We thank this Reviewer for his remarks and recommendation.

Author's Response to Reviewer #3:

The authors where able to address the concerns I expressed, specifically the issue of generality/utility of the presented results from additional experiments and conclusions.

Response: We thank this Reviewer for his/her remarks and recommendation.